

# Building a comprehensive library of observed Lagrangian trajectories for testing modeled cloud evolution, aerosol-cloud interactions, and marine cloud brightening

**Ehsan Erfani[1], Robert Wood[2], Peter Blossey[2], Sarah J. Doherty[2], Ryan Eastman[2]**

[1]Division of Atmospheric Sciences, Desert Research Institute, Reno, NV, USA

[2]Department of Atmospheric Sciences, University of Washington, Seattle, WA, USA

*Correspondence to:* Ehsan Erfani, (Ehsan.Erfani@dri.edu)

## Abstract

As marine low clouds' evolution is sensitive to the current state of the atmosphere and varying meteorological forcing, it is crucial to ascertain how cloud responses differ across a spectrum of those conditions. In this study, we introduce an innovative approach to

15 encompass a wide array of conditions prevalent in low marine cloud regions by creating a comprehensive library of observed environmental conditions. Using reanalysis and satellite data, over 2200 Lagrangian trajectories are generated within the stratocumulus deck region of the Northeast Pacific during summer 2018-2021. By using 8 important cloud-controlling factors (CCFs), we employ Principal Component Analysis (PCA) to reduce the dimensionality

of data. This technique demonstrates that two principal components capture 43% of the variability among CCFs. Notably, PCA facilitates the selection of a reduced number of trajectories (e.g., 54) that represent a diverse array of the observed CCF, aerosol, and cloud variability and co-variability. These trajectories can then be used for process model studies, e.g., with Large-Eddy Simulations (LES), to evaluate the efficacy of Marine Cloud Brightening.

Two distinct cases are selected to initiate two-day-long, high-resolution, large-domain LES experiments. The results highlight the ability of our LES to simulate observed conditions. Although perturbed aerosols delay cloud breakup and enhance cloud radiative effect, the strength of such effects is sensitive to "precipitation-aerosol feedback". The first case is



precipitating and shows the potential for "precipitation-driven" cloud breakup due to
positive precipitation-aerosol feedback. The second case is non-precipitating with classic
cloud breakup of "deepening-warming" type, highlighting the impact of entrainment.

## 1 Introduction

Marine stratocumulus (Sc) clouds are an important controller of climate because they cover
more than 20% of the ocean's surface, their albedo is much higher than that of the sea
surface, and their effect on outgoing longwave radiation is small (Wood, 2012). Changes in
the coverage or albedo of these clouds can therefore have significant impacts on Earth's
radiation budget, and biases in their representation in models can produce biases in
simulated climate. In addition, a large portion of the global climate forcing through aerosol-
cloud interactions (ACI) occurs in regions of extensive marine low clouds ( e.g. Carslaw et al.,
2013; Kooperman et al., 2012); accordingly, uncertainty in present-day anthropogenic
aerosol radiative forcing is largely attributed to uncertainty in aerosol indirect effects related
to low clouds (Forster et al., 2021; Sherwood et al., 2020).

Sc clouds have also been proposed as the potential target for the climate intervention
approach known as Marine Cloud Brightening (MCB), one of several methods of Solar
Radiation Modification (SRM) that have been suggested as a possible option for deliberately
reducing climate warming in the future. MCB would involve injecting sea salt particles from
sea water into the atmosphere in regions of marine low clouds. The idea is that these aerosols
would mix into the boundary layer air and up to the cloud base, where they would act as
cloud condensation nuclei (CCN), resulting in marine low clouds with a larger number of
small cloud droplets; this change would enhance cloud albedo (Twomey, 1977), increasing
sunlight reflection and cooling climate (Latham et al., 2012). Currently, it is thought that MCB
would be most effective when applied to regions of marine Sc clouds (Hill and Ming, 2012).
Early research on MCB shows it has the potential to cool the planet, yet significant
uncertainties still exist in predicting the efficacy of MCB within global climate models
(GCMs), because they do not resolve many of the complex physical processes associated with



both unperturbed marine low clouds and their interactions with aerosols (Wood et al., 2017).

Both the present-day effect of aerosols on climate through ACI and the MCB approach operate by changing the CCN population that is ingested into clouds. The initial response to increasing CCN, either with pollution aerosols or sea salt under MCB, is the Twomey effect (Twomey, 1977), which involves an enhancement of cloud droplet number concentration ($N_d$) and a resulting increase in cloud albedo if both cloud liquid water path (LWP) and cloud fraction (CF) are unchanged. The cloud "albedo susceptibility" to the Twomey effect is particularly sensitive to aerosol concentration, such that cloud albedo increases more significantly with lower aerosol concentration (i.e., in a clean environment than in an environment with already-elevated aerosol concentrations (Platnick and Twomey, 1994).

Importantly, in natural environments, neither LWP nor CF remains constant in clouds with altered. $N_d$. Early research indicated that smaller cloud droplets formed by enhanced aerosols diminish the efficiency of the collision-coalescence process, thereby suppressing precipitation and consequently increasing both cloud cover and cloud lifetime (Albrecht, 1989). More recent studies have demonstrated that ACI can include additional complex responses, such as increasing the entrainment of air at the cloud top and altering circulation in and adjacent to the cloud. Depending on the background environmental conditions (e.g., the strength of precipitation and entrainment), the cloud responses or "adjustments" in LWP and CF can act to either counteract or amplify the albedo increase produced by the Twomey effect (Glassmeier et al., 2021; Stevens and Feingold, 2009; Wood, 2021). However, the relative importance of different cloud adjustments on a global scale is still not fully understood (Christensen et al., 2022).

Despite the critical role marine low clouds play in climate, both through ACI in the present climate and for potential MCB, their accurate representation in global climate models continues to be a challenge (Lee et al., 2022; Stjern et al., 2018). The coarse resolution of these models is insufficient for directly simulating the cloud and aerosol processes that drive these clouds' evolution and their responses to aerosol perturbations, which necessitates parameterizations of these processes (Doherty et al., 2022; Erfani and Burls, 2019; Hannay



et al., 2009; Zelinka et al., 2017). Large-eddy simulation (LES), on the other hand, proves more effective since it is able to resolve numerous processes related to turbulence, aerosols, and clouds within the marine boundary layer or MBL (Wyant et al., 1997; Sandu and Stevens, 2011; Berner et al., 2013; Blossey et al., 2021; Yamaguchi et al., 2017). An objective of this
study is to establish an approach whereby the LES model's ability to represent the evolution of marine Sc clouds across a realistic range of background aerosol and meteorological conditions can be systematically tested. The goals of doing so are to advance our understanding of the factors controlling marine low clouds' contribution to Earth's radiative balance, their role in climate forcing by ACI, the potential for MCB to cool climate and affect
climate risks, and, ultimately, to be able to use LES simulations to test and improve representation of these clouds and both inadvertent (pollution) and intentional (MCB) impacts of aerosols on clouds in global-scale models.

The focus of this study is on regions dominated by Sc clouds and, importantly, includes broad variability in the factors that drive the evolution of these clouds with time. A distinctive
characteristic of marine low clouds over eastern oceans is the stratocumulus-to-cumulus transition (SCT), a change in cloud regime that occurs as lower tropospheric air masses in Sc-dominated regions move equatorward, carried by the trade winds. The first theory of what drives the SCT, termed "deepening-warming" (Bretherton and Wyant, 1997; Wyant et al., 1997), describes that the increased sea-surface temperatures (SST) experienced during
the equatorward movement of a well-mixed Sc-topped MBL cause a deepening and decoupling of the MBL that results in the formation of cumulus (Cu) clouds beneath Sc clouds. Simultaneously, the entrainment of dry air from the free troposphere (FT) intensifies at the top of the Sc layer and therefore leads to the dissipation of these clouds (Bretherton and Wyant, 1997; Sandu and Stevens, 2011; Wyant et al., 1997; Zhou et al., 2015). A more
recent theory, a "precipitation-driven" SCT (Yamaguchi et al., 2017), highlights the significant role of aerosols and a positive "precipitation-aerosol feedback", wherein enhanced precipitation results in a more efficient collision–coalescence process that effectively removes cloud droplets and aerosols from the Sc layer. This clean layer favors the formation of fewer, larger cloud droplets, which in turn intensifies precipitation (Yamaguchi
et al., 2017; Wood et al., 2018; Diamond et al., 2022; Erfani et al., 2022).



Data from an observational field campaign were used to both initialize and then test the fidelity of one LES model (System for Atmospheric Modeling or SAM; see Sect. 2.4) in simulating the SCT in the northeast Pacific (NEP) region (Blossey et al., 2021; Mohrmann et al., 2019). The Cloud System Evolution in the Trades (CSET) field campaign, took place over

the NEP in July and August 2015 (Albrecht et al., 2019). To study the movement of air masses, flights first sampled the MBL and lower FT offshore of California, and then the airmass was re-sampled two days later near Hawaii. Therefore, there are two aircraft intersects for each trajectory providing in-situ observations of cloud, aerosol, and meteorological properties. Erfani et al. (2022) selected two Lagrangian trajectories from the CSET campaign and

conducted a combination of low- and medium-domain-size LES model runs initialized using baseline and perturbed aerosol concentrations in the MBL and FT in order to explore the sensitivity of cloud evolution, including the SCT, to variations in aerosol concentrations and model domain size. The LES used in that study prognoses aerosol and cloud mass and number concentration; this adds more degrees of freedom, making it more challenging to

produce realistic simulations. Nonetheless, the LES did a better job of reproducing the evolution of the aerosol and cloud fields across the 3.5-day simulations in the first case (e.g., L06-Tr2.3) than the second case (e.g., L10-Tr6.0). The background environmental conditions differed between the two cases: the first case is clean and precipitating with an initially well-mixed Sc-topped MBL, and the second case is polluted and non-precipitating with an initially

decoupled MBL. As a result, the response of marine low clouds to aerosols and the strength and sign of cloud adjustments differ between the two cases.

The case studies presented by Erfani et al. (2022) inform the analysis presented here, which aims to provide a framework for a more systematic exploration of how aerosols affect low marine cloud evolution. Quantifying these effects requires a comprehensive understanding

of cloud responses under the full range of aerosol and meteorological conditions present in the eastern subtropical oceans. Here we present an approach for creating a comprehensive library of Lagrangian observations and meteorological forcings in order to represent a full spectrum of environmental conditions common in low marine cloud regions. We then apply Principal Component Analysis (PCA) to a range of cloud-controlling factors (CCFs) in order

to minimize the data dimensionality and to create a representative phase space of cloud





properties. This allows us to identify a subset of cases that are representative of the range of conditions that drive cloud evolution in a given region. In addition, we develop a methodology for routinely initializing and forcing detailed LES simulations with satellite and reanalysis data, rather than relying on aircraft measurements, which are only intermittently

available.

Building on Erfani et al. (2022), we perform LES runs with both baseline and perturbed aerosol concentrations for two cases identified from the sub-set of cases selected using the PCA method described above. These serve as examples to test the performance of our LES and to simulate low marine cloud evolution and ACI. A later study will use this same

approach to more comprehensively analyze LES model performance across an ensemble of simulations based on the subset of cases. The rest of this paper is organized as follows: Section 2 describes the observational data and model utilized in the study, along with the innovative statistical approach and design of the LES experiments. In Sect. 3, we explain the outcomes of the statistical analysis. The results of the LES experiments are examined in Sect.

4, a summary is provided in Sect. 5, and concluding remarks are given in Sect. 6.

## 2   Data and Methods

### 2.1   Data

We utilize a variety of reanalysis and satellite data for the Lagrangian study of Sc clouds.

Cloud and radiation properties such as CF, LWP, ice water path (IWP), cloud-top height (CTH), and the radiative fluxes were derived from the National Aeronautics and Space Administration (NASA) level 3 Clouds and the Earth's Radiant Energy System (CERES)—Synoptic top of the atmosphere (TOA) and surface fluxes and clouds (SYN) (Doelling et al., 2016). Additional sources of satellite-retrieved LWP include the Advanced Microwave

Scanning Radiometer (AMSR; Kawanishi et al., 2003), and the Special Sensor Microwave Imagers (SSMI; Wentz et al., 2012). Furthermore, we use CTH estimated from the NASA Moderate Resolution Imaging Spectroradiometer (MODIS) cloud top temperatures, tuned



with cloud top heights from the Cloud-Aerosol Lidar and Infrared Pathfinder Satellite Observations (CALIPSO, Vaughan et al., 2004) using the algorithm described in Eastman et al., (2017). Estimated warm rain rates are derived from AMSR 89 GHz brightness temperatures tuned using concurrent CloudSat Rain profile rain rates (Lebsock and L'Ecuyer, 2011) for marine low clouds, which is available twice daily (Eastman et al., 2019). The European Center for Medium-Range Weather Forecasts (ECMWF) reanalysis version 5 (ERA5) data is used for obtaining meteorological conditions and cloud properties, such as temperature ($T$), water vapor mixing ratio ($q$), horizontal wind speed (WS), vertical velocity in pressure coordinates ($\omega$), CF, and LWP (Hersbach et al., 2020). Estimated Inversion Strength (EIS) is then calculated following the Wood and Bretherton (2006) derivation, as an index to measure MBL static stability. The mass mixing ratio of aerosol species is

Table 1. A summary of datasets used in this study.

| Dataset | ERA5 (surface & pressure levels) | MERRA2 M2I3NVAER (aerosol variables) | CERES SYN L3 (radiation/ cloud variables) | SSMI V08 L3 | AMSR-2 V08 L3 | AMSR-2 V08 L3 | MODIS |
|---|---|---|---|---|---|---|---|
| Important Variables | WS, $P$, $T$, $q$, $\omega$, EIS, SST, $Z_{inv}$, $w_e$, CF, LWP | $N_a$ | CF, LWP, CTH, $N_d$, $r_e$, $\tau_c$, OLR, SW CRE | LWP | LWP | rain rate | CTH |
| Reference | Hersbach et al. (2020) | Gelaro et al. (2017) | Doelling et al. (2016) | Wentz et al. (2012) | Kawanishi et al. (2003) | Eastman et al. (2019) | Eastman et al. (2017) |
| Temporal Resolution | Hourly | 3-hourly | Hourly | Two times per day | Two times per day | Two times per day | 01:30 LT, 13:30 LT |
| Spatial Resolution | 0.25×0.25° | 0.5×0.625° | 1×1° | 0.25×0.25° | 0.25×0.25° | 0.25×0.25° | 1×1° |
| Vertical Levels | 37 | 72 | --- | --- | --- | --- | --- |

extracted from the NASA Modern-Era Retrospective analysis for Research and Applications, Version 2 (MERRA2; Gelaro et al., 2017) reanalysis. The MERRA2 reanalysis dataset is generated using the Goddard Chemistry Aerosol Radiation and Transport (GOCART) model, run with assimilated satellite retrievals and meteorological data. Accumulation-mode



aerosol number concentration ($N_a$) is calculated from the MERRA-2 mass mixing ratio of aerosol species, with assumed particle size distributions following a methodology described in Appendix A of Erfani et al. (2022). A summary of the datasets used in this study is provided in Table 1.

### 2.2   Airmass trajectories

The target region for this study encompasses the Sc cloud deck in the NEP. Six initial locations in this region (see Fig. 1a) are selected as the starting points of forward airmass trajectories that are then used to derive representative cloud properties in this region. All initial locations, except "North", are based on their use in previous studies. The Sandu2010 location is selected following Sandu et al. (2010) who analyzed numerous trajectories from this location. The GPCI S9-S12 locations are part of an enhanced observational field campaign, called Global Energy and Water Cycle EXperiment (GEWEX) Cloud Systems Study (GCSS) Pacific Cross-section Intercomparison (GPCI) (Lewis et al., 2012).

We employ a trajectory code developed at the University of Washington (UW) (Bretherton et al., 2010; Eastman and Wood, 2016) to generate a total of 2208 Lagrangian isobaric (950 hPa) forward trajectories using ERA5 wind data. These trajectories cover a timespan of 86 hours (3.5 days) and are all from the summer months (June, July, and August, or JJA) in the years 2018-2021. To compile meteorological, cloud, radiation, and aerosol properties along the trajectories, we utilize the "uw-trajectory" Python package which was originally developed for the CSET field campaign (Mohrmann et al., 2019) but has since been modified for use with any initial time and location, provided the necessary datasets are available. The uw-trajectory package provides data averaged over a 2° × 2° box centered on each trajectory point at each time, using approximately the same sample sizes as Eastman and Wood (2016). Although the trajectory files are created for a period of 86 hours, only the first 48 hours of trajectories are used for data analysis and modeling in this study. The spread for each





**a)**

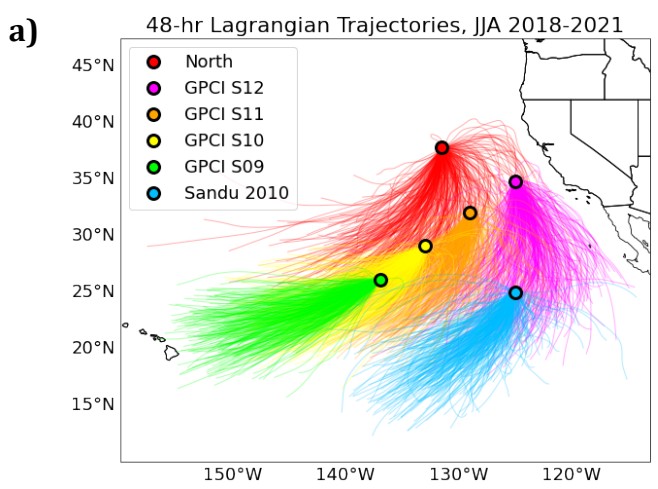

**b)**

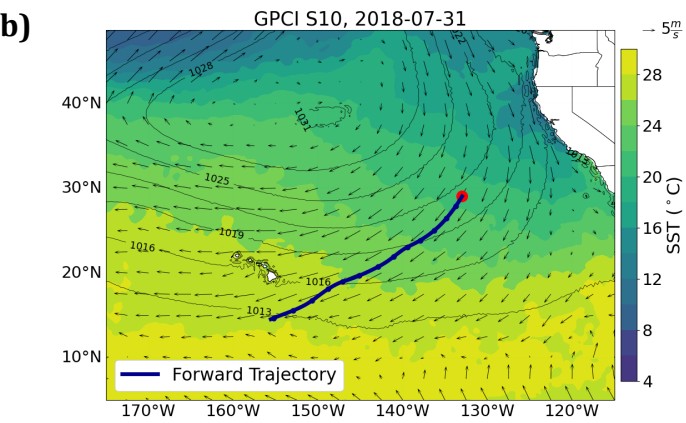

**c)**

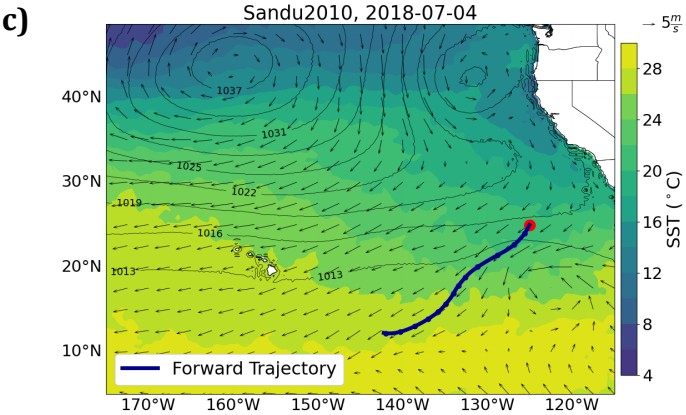

Figure 1. (a) Lagrangian isobaric (950 hPa) 48-hour forward trajectories initialized over six select locations in the stratocumulus deck region of the Northeast Pacific for every day during June-August 2018-2021. Excluded are the 4% of the trajectories that pass close to the coast or over land. See Sect. 2.2 for a description of initial locations. (b) and (c) Two Lagrangian trajectories (dark blue solid lines) used here as case studies for LES modeling. The shaded contours, black contours, and vectors show the ERA5 sea surface temperature, mean-sea-level pressure, and 10 m wind vectors, respectively, averaged for a 48-hour period starting from the initial time of the trajectory.



variable is defined as the standard deviation within that box. As we are interested in studying influences on the SCT, we exclude trajectories that pass close to the coastlines or over land

(4% of the total). Trajectories with significant ice content, i.e., those with an IWP exceeding 50 mg g$^{-1}$ lasting more than three hours during a 48-hour trajectory, are also excluded. This reduces the total number of trajectories to 1663.

## 2.3 Principal Component Analysis (PCA)

The goal of using PCA in this study is to be able to identify a set of airmass trajectories (cases)

for which detailed LES simulations can be run, both to test the fidelity of the LES in simulating cloud evolution and then to study how cloud evolution is affected by aerosol perturbations. To make the most effective use of available computing resources, we would like to simulate a small subset of the >1000 trajectories described above. However, that subset should encompass the phase space of observed CCFs and cloud properties so that, collectively, they

represent the range of variability in cloud evolution in the selected region. This is done by identifying a reduced set of principal components (PCs) that can explain a large fraction of the variability in our selected CCFs.

PCA is a statistical technique used to identify patterns in data and reduce its dimensionality by transforming a dataset with $n$ physical variables to $n$ variables in the variance space, called

principal components (PCs). PCA is an optimal technique to explain variations because it starts by projecting data onto the direction of the largest variance and then repeats this for an axis with the second largest variance. This process is continued to the $n^{th}$ axis of variance. For this reason, PC1 represents the largest variance, PC2 the second largest variance, and so on. An important benefit of PCA is that many variables can be represented by the first few

PCs because they often explain the majority of total variance, and therefore PCA reduces the dimensionality. Another benefit of PCA is that it removes co-variability by producing PCs that are orthogonal and uncorrelated (Hartmann, 2008).

Mathematically, PCA is expressed for each PC by decomposing a symmetric matrix as: $\mathbf{\Gamma}\boldsymbol{\alpha} = \lambda\boldsymbol{\alpha}$, where $\boldsymbol{\alpha}$ is eigenvector, $\lambda$ is eigenvalue, and $\mathbf{\Gamma}$ is the covariance matrix with each of its

elements showing the covariance between two variables. For each PC, the eigenvalue





describes the percentage of variance explained by that PC and each eigenvector element shows the importance of an input variable for that PC (Wei, 2018).

Before applying PCA, we standardize each variable ($V$) by calculating $V_{standardized}$ from $V$, mean value ($\bar{V}$) and the variable's standard deviation ($\sigma_V$) as: $V_{standardized} = \frac{(V-\bar{V})}{\sigma_V}$. This

standardization is necessary to ensure that a variable with a wide range does not dominate the PCA. PCA is sensitive to outliers and cannot be performed if there are missing values in the datasets; however, these issues do not exist in the datasets used in this study.

A single PC analysis is conducted for all 1663 trajectories in the study region based on eight variables: the along-trajectory means and the differences between the beginning and end of

each trajectory for the four CCFs: EIS, 700-hPa $q$, 700-hPa $\omega$, and 10-m WS, where the 700-hPa vertical level is used to represent the lower FT. These CCFs, along with SST, and mean-sea-level pressure ($P_{MSL}$), have been shown to be the most important CCFs for the development of marine low clouds (Klein et al., 2017; and references therein). Here we excluded SST and $P_{MSL}$ from PCA because they have high co-variability with other CCFs, but

a lower correlation coefficient (R-value) with our cloud properties of interest. For example, the R between SST and EIS is -0.6, and $\Delta$SST and $\Delta P_{MSL}$ are highly correlated with WS$_{10m}$ (0.6, -0.5, respectively; See Fig. S1). To gain insight into what factors most strongly affect cloud evolution along the SCT, we focus on the percentage of variance explained by each PC, and which variables within the leading PCs (as given by the R between each PC and a given

physical variable) contribute the most to the variability within these PCs.

To indicate the robustness of correlations, the probability, or p-value, is determined using the t-test (or t-distribution, $t_d$) for the statistical significance of the correlation, following Lowry (2014): $t_d = \frac{R}{\sqrt{\frac{1-R^2}{df}}}$, where $df = N^* - 2$ is the degrees of freedom. Here, $N^*$ is the independent number of datapoints, which is smaller than the total number of datapoints ($N$,

which is equal to 1663 in this study), because synoptic variability exists on a multi-day scale. Following Hartmann (2008), $N^*$ is calculated as: $N^* = \frac{N\Delta t}{2t_e}$, where $\Delta t$ is the time interval between two datapoints (equal to 1 day in this study), and $t_e$ is the time interval during




which the autocorrelation becomes smaller than $e^{-1}$. By calculating autocorrelation for each of the 6 locations and each of the 4 years (figure not shown), it is seen that $t_e$ generally does not exceed 4 days. This results in a value of 208 for $N^*$, which then gives a $t_d$ value of 2.03 and a p-value of 0.05, when the R-value is equal to 0.14. This means that an R-value of 0.14 or higher is statistically significant at a confidence level of 95% (or when the p-value is lower than 0.05 for non-directional conditions) for our specific dataset.

## 2.4   Model

We perform LES modeling using SAM (Khairoutdinov and Randall, 2003), version 6.10.9, with the goal of producing detailed, high-resolution, large-domain simulations of cloud evolution. In previous work at the University of Washington, SAM was coupled to a single-mode, bulk, log-normal, two-moment aerosol scheme (Berner et al., 2013). This scheme prognoses the mass and number concentration of accumulation mode aerosols in the boundary layer by computing budget tendencies due to accretion or coalescence scavenging, interstitial scavenging, autoconversion, activation, sedimentation, surface processes, and entrainment from the FT for dry aerosol (unactivated), in-cloud droplets, and raindrops. A detailed calculation of each tendency term is described by Berner et al. (2013). Warm-cloud microphysics in SAM uses the Morrison parameterization (Morrison et al., 2005), which is a bulk double-moment scheme that predicts cloud droplet and raindrop number concentrations with gamma distributions and parameterizes activation of cloud droplets from two modes of aerosols based on Abdul-Razzak and Ghan (2000). The ice phase is turned off in the microphysics parameterization of our LES. Additionally, we use the Rapid Radiative Transfer Model for Global Climate Models (RRTMG) (Mlawer et al., 1997), and cloud optical parameterizations from the Community Atmosphere Model version 5 (CAM5) as described in Neale et al. (2010).

The simulations in this study are similar to those in Erfani et al. (2022), with three main differences. First, the LES is initialized and forced using satellite and reanalysis meteorological and aerosol data in order to test the fidelity of LES in the absence of aircraft measurements, which are not widely available over remote oceans. Second, the simulations are initialized using *sharpened* profiles of temperature and moisture. This is done to



overcome the fact that the ERA5 thermodynamic profiles do not well represent the structure of the inversion layer when compared to aircraft measurements (i.e. see Figs. 4 and 8 in Erfani et al., 2022).

The profile sharpening procedure is explained in Appendix A in detail but is summarized here: The procedure uses the ERA5 $T$ and total water mixing ratio ($q_t$) profiles and the microwave LWP as inputs. The MBL inversion height ($Z_{inv}$) is calculated from the ERA5 profiles, and the FT profiles are extrapolated from 500m above $Z_{inv}$ down to $Z_{inv}$. The MBL profiles are then adjusted based on minimizing an error function that optimizes LWP in the

adjusted profile against the microwave LWP while preserving the vertical integrals of the ERA5 density temperature ($T_\rho$) (defined in Appendix A) and $q_t$. This sharpened profile is then used to initialize the LES runs.

Table 2. A summary of large-eddy simulation runs conducted in this study. Note that two separate Lagrangian

trajectories are selected, and for each of them, 3 runs are conducted.

| Trajectory | Run name | Initial MBL $N_a$ | Description | Initial time (Z) | FT $N_a$ | Run time (h) | Horizontal resolution (m) | Domain size (km) | Vertical level # |
|---|---|---|---|---|---|---|---|---|---|
| **GPCI S10 (2018-07-31)** | **ctrl** | MERRA | Baseline run: initialized with MERRA MBL $N_a$ | 09 | MERRA | 48 | 100×100 | 51.2×51.2 | 260 |
| | $N_a\times3$ | MERRA×3 | Run initialized with MERRA MBL $N_a$ multiplied by 3 | | | | | | |
| | $N_a\times9$ | MERRA×9 | Run initialized with MERRA MBL $N_a$ multiplied by 9 | | | | | | |
| **Sandu 2010 (2018-07-04)** | **ctrl** | MERRA | Baseline run: initialized with MERRA MBL $N_a$ | 09 | MERRA | 48 | 100×100 | 51.2×51.2 | 260 |
| | $N_a\times3$ | MERRA×3 | Run initialized with MERRA MBL $N_a$ multiplied by 3 | | | | | | |
| | $N_a\times9$ | MERRA×9 | Run initialized with MERRA MBL $N_a$ multiplied by 9 | | | | | | |



The third difference between this study and that of Erfani et al. (2022) is that here we conduct two stages for each simulation: startup and active. The 8-hour startup stage serves as the spinup period and is forced with meteorological conditions and aerosol properties that are constant in time, using instantaneous profiles from the initial time of each trajectory (e.g., 09Z). The startup stage is run for nighttime-only conditions and facilitates the development of mesoscale cells. The sharpened temperature and moisture profiles are used in this stage. Other forcing fields are ERA5 $\omega$, geostrophic winds, SST (Figs. 1b and 1c), and large-scale horizontal advection of temperature and moisture. To enable the formation of an Sc-topped, well-mixed MBL in the LES, a nudging time scale of 1 hour is selected for profiles of $T$ and $q_t$ within the MBL during this startup stage. This stage is important for creating a steady geostrophic wind throughout the spinup period; without this stage, transients in the winds can occur and cause wind and surface flux errors during the early part of the simulation, as evident in previous LES studies of the CSET campaign (e.g., Blossey et al., 2021; Erfani et al., 2022). A 48-hour active stage is then branched from the startup stage and serves as the main run with realistic forcings that change over time. In this stage, the nudging of aerosols, temperature, and moisture within the MBL are turned off to allow for the natural development of aerosols and clouds within the MBL. During both the startup and active stages the FT profiles of aerosols, temperature, and moisture are nudged to the ERA5 reanalysis values with a time scale of 1 hour. For nudging winds, a longer time scale of 12 hours is selected.

Table 2 summaries the experiments in this study. The number of vertical levels in the model is 260, with the smallest vertical grid spacing being 7 m in the MBL top and Sc layer (from 450 m to 1200 m). Above and below this vertical range, the vertical grid spacing gradually expands, such that it is 167 m just below the model top (which is at 4800 m) and is 20 m immediately above the ocean surface. The horizontal resolution is 100 m × 100 m and the horizontal domain size is 51.2 km × 51.2 km.

Three runs are conducted for each trajectory: ctrl, $N_a$ ×3, and $N_a$ ×9. All three runs are initialized and forced in the FT with MERRA2 time-varying aerosol profiles. The ctrl run is initialized in the MBL with MERRA2 aerosol profiles. As in Erfani et al. (2022) to test for the



sensitivity of cloud evolution to aerosols the $N_a \times 3$ and $N_a \times 9$ runs are initialized with MERRA2 MBL aerosol concentration multiplied by 3 and 9, respectively. Although simulated FT $N_a$ is nudged to MERRA2 $N_a$ throughout the simulation, the MBL $N_a$ is prognosed freely for a natural simulation of aerosols, clouds, and precipitation. The time-varying vertical profiles of MERRA2 $N_a$ are shown in Fig. 2 for two example trajectories and are derived from the mass mixing ratios of aerosol species, following Appendix A in Erfani et al. (2022).

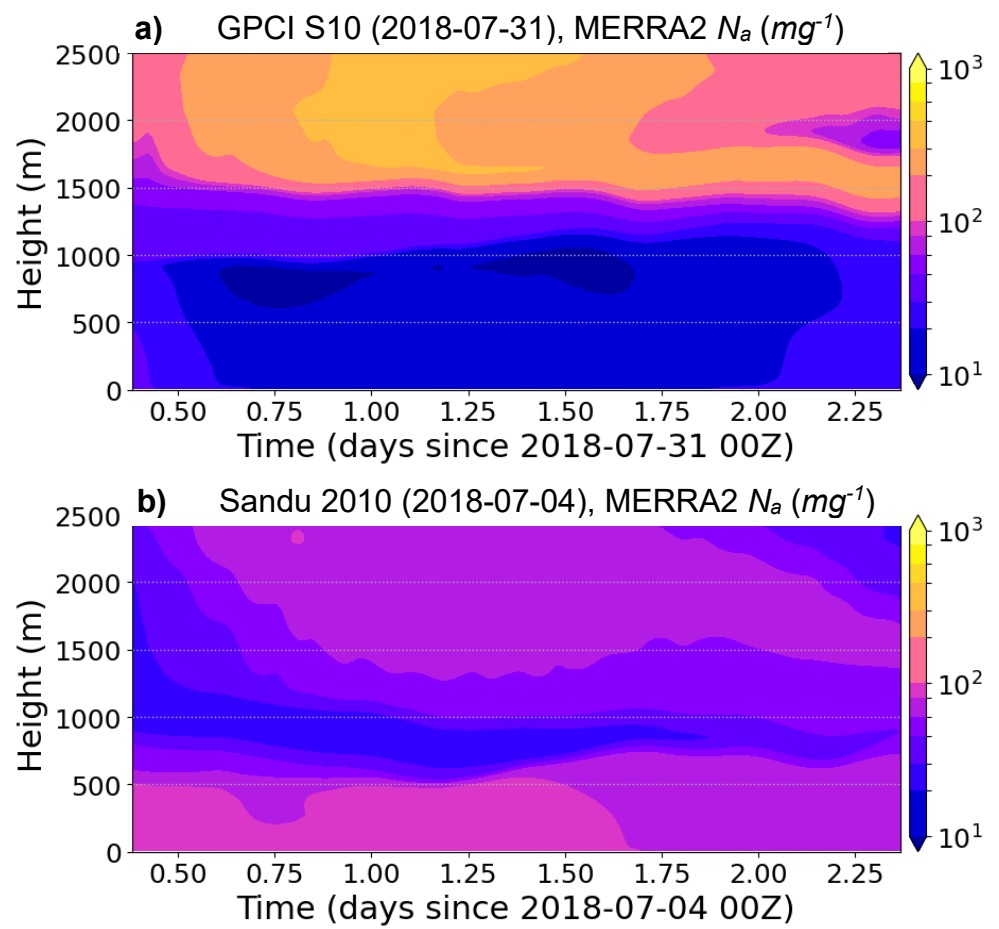

Figure 2. The time-height plot of MERRA2 $N_a$ for the two trajectories used in this study for the LES case studies.



## 3 Statistical analysis

### 3.1 PCA results

One objective of using PCA in our study is to determine how the main modes of variation in key cloud properties are related to a select set of CCFs. This approach helps simplify the complex interactions between clouds and their environment by focusing on the most significant modes of variability and identifying the combinations of factors driving this variability. We first explore the R-values between various cloud properties and CCFs, as shown in Fig. S1. For instance, the R between CF and EIS is 0.31, which indicates a moderate positive relationship. In addition, R between $N_d$ and EIS is 0.36, which suggests that an increase in EIS is associated with an increase in $N_d$. Stronger stability near the inversion layer leads to weaker mixing, inhibiting the MBL from deepening, which ultimately enhances humidity and clouds near the top of the MBL (Klein et al., 2017; Wood and Bretherton, 2006). FT subsidence, represented by 700-hPa $\omega$, is correlated with cloud properties (e.g., R between $\omega$ and CERES CTH is equal to 0.22, and R between $\omega$ and CF is equal to -0.13), because weaker FT subsidence leads to MBL deepening and increased cloudiness (Klein et al., 2017; Myers and Norris, 2013). Surface WS is correlated with CTH, LWP, and precipitation with the highest R-values corresponding to the changes in these properties along the trajectory: CERES $\Delta$CTH (0.38), $\Delta$log(SSMI LWP) (0.28), and $\Delta$log(precip) (0.25). This is due to enhanced latent heat fluxes from the ocean surface that intensify latent heat release and facilitate cloud formation (Bretherton et al., 2013; Brueck et al., 2015). As explained in Sect. 2.3, absolute values of R greater than 0.14 are considered statistically significant, which provides confidence in the relationships seen between these variables.

Figure 3a illustrates the contribution of each PC to the total variance by showing the percentage of variance explained by each PC, when PCA is conducted with 8 variables as inputs. This visualization helps us to both understand the distribution of variance across the PCs, and determine the number of PCs necessary to capture sufficient variability in our dataset. Since the PCs collectively encompass the entire variance in the dataset, the summation of the percentage of variance for all 8 PCs amounts to 100%. Notably, the first PC explains 24% of the total variance, while the second PC explains an additional 19%.




Together, these two PCs account for a significant portion of the variability in the data. Thus,

43% of the information regarding the variation in CCF properties is captured within PC1 and

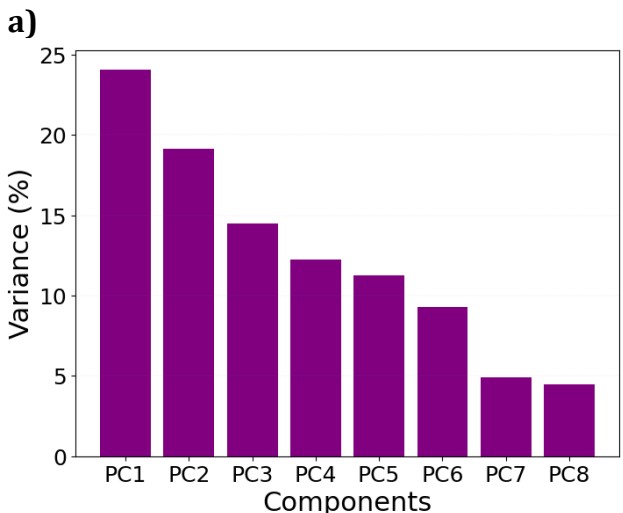

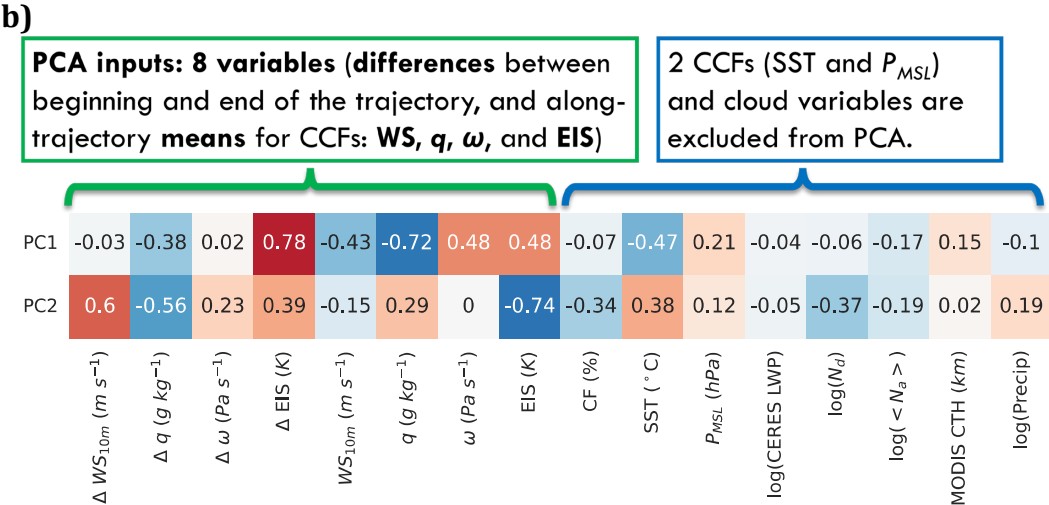

Figure 3. (a) The result of Principal Component Analysis (PCA) showing the percentage of variance explained by each Principal Component (PC). (b) The relationship between the two PCs and key meteorological conditions and cloud properties are quantified through their correlation coefficient (R-value). The inputs to the PCA are presented as the first 8 variables on the left; these are the differences between the beginning and end of the trajectory and the along-trajectory means for the cloud controlling factors (CCFs): WS, $q$, $\omega$, and EIS.



PC2, highlighting their importance in understanding cloud formation and evolution. Note that the contributions to the total variance differ among all the PCs, which shows different levels of importance among PCs.

In order to provide insight into which variables are most strongly associated with the modes of variation captured by PC1 and PC2, the R-values between each of the first two PCs and important CCFs, meteorological conditions, and cloud properties are shown in Fig. 3b. A more comprehensive set of R-values is provided in Fig. S1. For each PC, the R-values for input variables (i.e., the first 8 variables in each row) are associated with the eigenvector for that PC and determine the contribution of the input variable to that PC. This helps identify which CCFs are the most influential in the modes of variation represented by PC1 and PC2. The highest R with PC1 is for ΔEIS (0.78) and FT $q$ (-0.72); i.e., the change in EIS along the trajectory and FT $q$ are the most significant properties driving variability in PC1. For PC2, EIS and ΔWS are the most important variables, with R-values of -0.74 and 0.6, respectively, highlighting their roles in the mode of variation represented by PC2. Although SST is not an input to the PCA, it is correlated with both PC1 (-0.47) and PC2 (0.38), as SST is an input when computing EIS (Fig. S1). Additionally, PC1 and PC2 explain variations in some key cloud properties, as indicated by the fact that PC2 has R-values of -0.34 and -0.37 with CF and $N_d$, respectively, and PC1 has R-values of -0.28, -0.28, and -0.22 with CERES ΔCTH, Δlog(precip), and SSMI Δlog(LWP), respectively. This means that associations of the CCFs with cloud properties and their evolution along these Lagrangian trajectories are detected by PCA.

## 3.2 Phase space

To determine the full phase space of cloud variability in this region and identify a reduced number of trajectories that represent this range of variability, PC1 vs. PC2 is plotted for all qualifying trajectories (Fig. 4a). Additionally, and for each of the six select locations identified in Fig. 1a, 9 trajectories are selected that represent the values of (-1.5σ, 0, 1.5σ) in the 2-D PC1-PC2 plane (where σ is the standard deviation of each individual PC). This approach allows us to capture a representative sample of trajectories that encompass the range of



variability in the PCs phase space. As a result, the variability across 1663 trajectories can be sampled by just the 54 trajectories, shown with colored markers in Fig. 4a.

The selection of 9 points for each initial location is intended to represent reasonable variations associated with each initial location, as 87% of data points in a normal distribution

fall within 1.5 standard deviations. It is noteworthy that the 9 points for each initial location correspond to different parts of the PCs plane, which hints at the distinct characteristics of each initial location. For instance, data points for the "North" initial location (see Fig. 1a) tend to be more frequent at larger values of PC1 and PC2, whereas the frequency of data

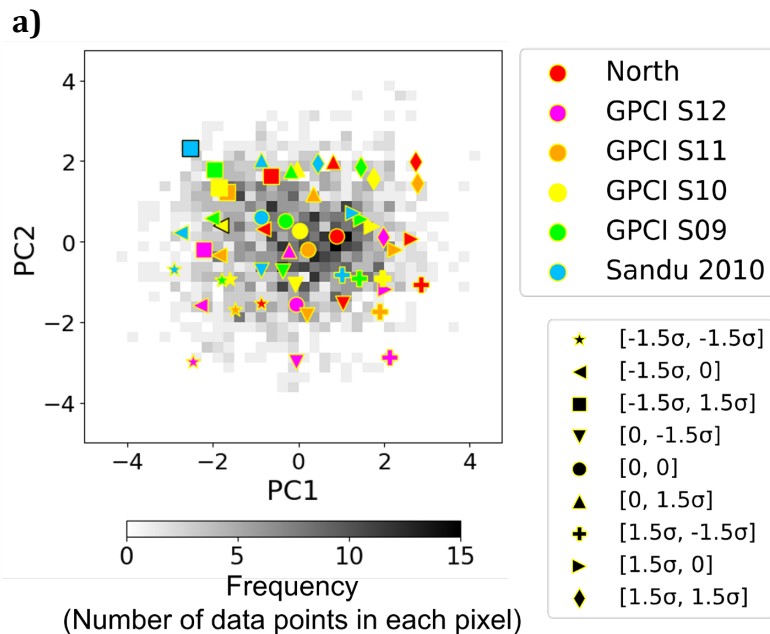

Figure 4. Phase space of variables. a) Frequency plot of the first two principal components (PCs) for all qualifying trajectories used in the PCA. Here, each trajectory is represented as one data point. The grey shades show the frequency, i.e., the number of trajectories in each pixel. PCs associated with the six select locations shown in Fig. 1a are indicated as colored markers. For each location, different marker shapes are used to show

nine trajectories that correspond to the standard deviation (-1.5σ, 0, 1.5σ) in this 2D space. b) Each panel shows a frequency plot for pairs of cloud-controlling factors and cloud variables averaged along the trajectories, with their correlation coefficient shown in the box. The markers in each panel show the 54 points selected from the PC1-PC2 space mapped to the space for that pair of variables. In each panel, the two markers with black edge color show the two cases used for LES modeling in Sect. 4.



**b)** 440



Figure 4. Continued.



points for "GPCI S12", the closest initial location to the coast, is greater for lower values of PC1 and PC2. This shows that the PCA is sensitive to the geographical location of the trajectory origin, likely because the variability and co-variability of CCFs recognized by PCA occur both in space and time.

To assess the impact of reducing the number of trajectories from the full set on coverage of the range in the physical variables of interest, the data points in the PC1-PC2 plane are mapped to the corresponding CCFs and cloud properties and, as in Fig. 4b, both the values for all trajectories (grayscale symbols) and for just the 54 select trajectories (colored symbols) are shown. Also as in Fig. 4b, for each pair of variables, the frequency of the along-trajectory averages in PC space is conveyed by the grayscale. Fig. S2 shows the same but for the differences (change) in the CCFs and cloud properties between the beginning and end of the trajectories. This analysis summarizes the changes in variables over the course of the trajectories and their potential implications for cloud development.

While the distribution of data points varies significantly across the different phase planes in Fig. 4b, the 54 selected data points successfully represent much of the full spectrum of CCFs and cloud variables in each panel, indicating that this reduced set of trajectories effectively captures the key patterns and variations in the datasets.

## 4   Numerical modeling

Here we take two of the 54 selected trajectories identified as covering the range in cloud variability at our six representative sites in the NEP region and use them to demonstrate our approach to testing the LES-simulated cloud evolution against observed cloud evolution starting in the Sc region and moving toward the more Cu-dominated region. In a later study, this approach will be used to statistically analyze model performance across all 54 cases and, informed by this baseline of model performance, to systematically study the response of clouds to aerosol perturbations across all 54 cases.



## 4.1 First case: Trajectory GPCI S10 (2018-07-31)

### 4.1.1 Observed characteristics

During the two-day period of this trajectory, a permanent subtropical high-pressure system was located over the NEP, producing northeasterly surface winds in its southeastern flank and along the trajectory (Fig. 1b). Based on phase space analysis for satellite and reanalysis data (Fig. 4a), this case is characterized by an average PC2 value and a negative PC1 value. Among the 54 cases selected by PCA, and considering along-trajectory averages, it exhibits very strong 10-m WS, very weak $\omega$, nearly overcast conditions ($\sim$ 90%), and strong LWP (Fig. 4b).

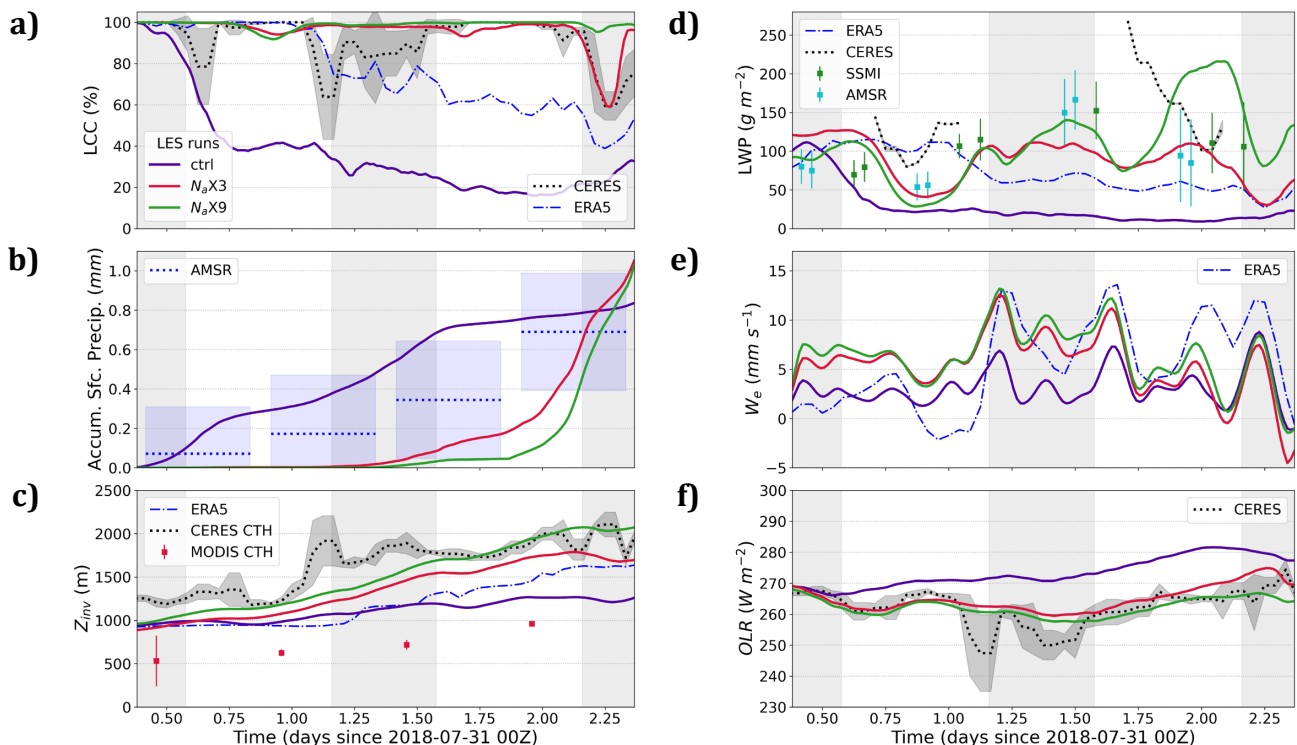

Figure 5. Time series of various observed and simulated domain-averaged meteorological variables for the GPCI S10 (2018-07-31) trajectory. (a) low cloud cover (LCC), (b) accumulated surface precipitation, (c) inversion height ($Z_{inv}$), (d) liquid water path (LWP), (e) entrainment rate ($w_e$), and (f) outgoing longwave radiation (OLR). The nighttime periods are indicated with light gray background shading.





Figures 5 and 6 show time series of various meteorological and aerosol properties along the
trajectory from different observationally-based datasets as well as from the LES runs.
According to the CERES low cloud cover (LCC) retrievals (Fig. 5a), cloud breakup, defined as
the reduction of domain-averaged LCC to 50%, does not occur throughout the 48-hour
period, and overcast conditions prevail for the majority of the time. The ERA5 reanalysis LCC

is lower than CERES LCC, as it gradually starts decreasing on the evening of the first day. A
comprehensive comparison of MODIS and ERA5 cloud cover on a global scale and for the
NEP region shows that ERA5 cloud cover is biased low (Wu et al., 2023). Considering that

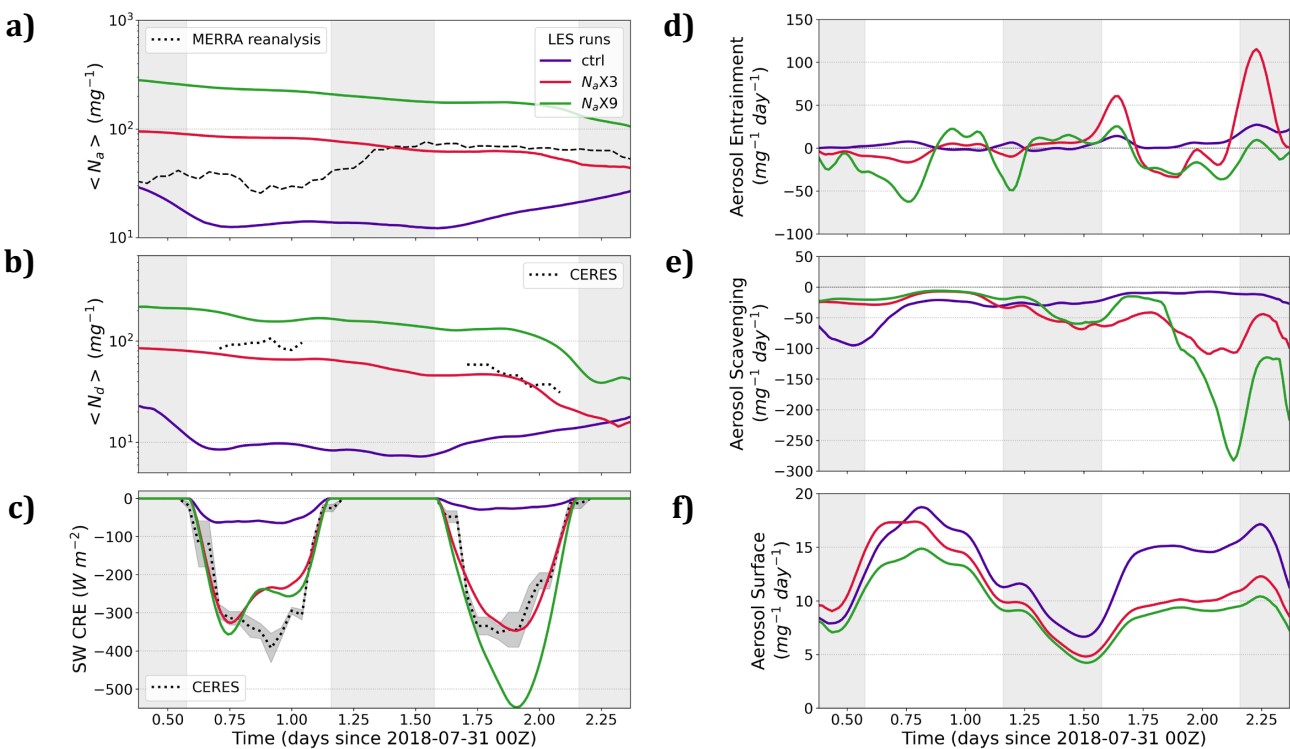

Figure 6. Time series of various observed and simulated domain-averaged variables for the GPCI S10 (2018-
07-31) trajectory. (a) total aerosol number concentration averaged within MBL ($<N_a>$), (b) cloud droplet
number concentration averaged within MBL ($<N_d>$), (c) shortwave cloud radiative effect (SW CRE) at the top
of atmosphere (TOA), (d) aerosol entrainment from the FT, (e) aerosol scavenging averaged within MBL, and
(f) aerosol surface fluxes. The nighttime periods are indicated with light gray background shading.



CERES data is based on MODIS retrievals, it is justified to consider CERES, and not ERA5, as ground truth in this study. This case features moderate precipitation (Fig. 5b), with a slight enhancement of precipitation in the second half of the trajectory, consistent with a gradual increase in cloud droplet effective radius ($r_e$), from approximately 13 to 16 μm (Fig. S3b). The observed LWP consistently remains above 50 g m$^{-2}$, with the AMSR-retrieved LWP reaching 170 g m$^{-2}$ towards the end of the second night (Fig. 5d). The observed microwave products (AMSR and SSMI) agree quite well, but CERES LWP is larger than the microwave-retrieved LWP at almost all times. (Note that we discard CERES LWP, $N_d$, $r_e$, and cloud optical depth, or $\tau_c$, when the zenith angle is greater than 70° to avoid erroneous values). The ERA5 reanalysis LWP underestimates the observed LWP in the second half of the trajectory, associated with a consistent reduction in ERA5 LCC during this period.

The observed CERES and MODIS CTH and ERA5 $Z_{inv}$ all show a gradual increase of approximately 500 m over the 48-hour period (Fig. 5c); however, there are differences between datasets, with MODIS CTH being the lowest and CERES CTH being the highest. This discrepancy is present in some other trajectories and is worth further investigation in future studies. In this case, the Cloud-Aerosol Lidar and Infrared Pathfinder Satellite Observations (CALIPSO) $Z_{inv}$ aligns more closely with ERA5 $Z_{inv}$ (figure not shown).

The MBL-averaged total aerosol number concentration, $<N_a>$, of about 30 mg$^{-1}$ from MERRA2 (Fig. 6a) indicates that the MBL is clean on the first day, but then $<N_a>$ more than doubles during the night likely due to entrainment from an FT with high $N_a$ (Fig. 2a). On the other hand, CERES $N_d$ is around 100 mg$^{-1}$ on the first day and decreases to around 30 mg$^{-1}$ by the afternoon of the second day (Fig. 6b). This implies that MERRA2 $<N_a>$ is biased low (by approximately a factor of 1/3) on the first day and is slightly biased high on the second day. Indeed, Erfani et al. (2022) showed that MERRA2 $N_a$ is biased low for higher $N_a$ values when compared to in-situ measurements for 53 Lagrangian trajectories during CSET campaign, and this highlights a limitation of MERRA2 reanalysis data in computing aerosol concentrations. Based on this, for future studies, we plan to initialize $N_a$ within the MBL based on CERES $N_d$ under the assumption that this $N_a$ estimate will be less biased than the MERRA2 reanalysis in regions of overcast cloud. For this trajectory, the changes in cloud





radiative properties from the decrease in $N_d$ and the increase in LWP from the first day to
the second day seem to cancel each other out, as the CERES retrieved $\tau_c$ and the TOA
shortwave cloud radiative effect (SW CRE) do not show significant day-to-day variations
(Figs. S3a and 6c).

### 4.1.2  Reference run ($N_a \times 3$)

Because the run initialized with MERRA2 $N_a$ within the MBL simulates early cloud breakup
in contradiction to the observations, the $N_a \times 3$ run is chosen as the reference simulation for
this case study. Important cloud properties, in particular LCC, LWP, $N_d$, precipitation, SW
CRE, and OLR, compare best to observed properties in the $N_a \times 3$ run (Figs. 5 and 6). This
highlights the ability of our LES to accurately simulate a range of cloud properties when it is
initialized with $N_a$ values that result in more accurate $N_d$, as the simulated $<N_d>$ along the
$N_a \times 3$ run trajectory is quite similar to the CERES retrieved $N_d$ (Fig. 6b). Previous studies
showed that $N_d$ scales with $N_a$ (Pringle et al., 2009; Svensmark et al., 2024). Both modeled
and CERES-retrieved LCC broadly agree, though the $N_a \times 3$ run is unable to simulate the
timing and strength of two brief episodes of LCC reduction seen in the CERES retrievals on
the first day and the following night (Fig. 5a). In the $N_a \times 3$ run, any LCC reductions from the
overcast conditions are associated with a remarkable decrease in LWP.

The $N_a \times 3$ accumulated precipitation is always less than mean AMSR precipitation, except on
the last night, but it generally stays within 1 standard deviation of observations (Fig. 5b).
Precipitation onset occurs in the middle of the second night (much later than observations),
enhances 12 hours later, and continues until the end of the simulation. This likely causes the
brief cloud reduction in the third night (Fig. 5a), along with the inhibition of $Z_{inv}$ growth (Fig.
5c).

$Z_{inv}$ in the $N_a \times 3$ run and in ERA5 are initially equal due to the nudging in the startup stage,
but $Z_{inv}$ in the $N_a \times 3$ run grows faster in the first half of the simulation and slower in the
second half than in the ERA5 dataset, ultimately remaining very close to ERA5 $Z_{inv}$ near the
end of the run. This also explains the stronger entrainment rate ($w_e$) in the $N_a \times 3$ run than in
ERA5 in the first half of the simulation and vice versa in the second half (Fig. 5e) (note that



$w_e$ is estimated as the tendency of $Z_{inv}$ relative to the $W$ at the inversion level; Blossey et al., 2021; Erfani et al., 2022).

The gradual reduction in $N_a$ and $N_d$ is due to a general strengthening of the aerosol
scavenging sink term (which is the sum of the accretion, autoconversion, and interstitial scavenging terms). The combined sink is stronger than the sum of the aerosol surface flux source term and entrainment from the FT (where the latter is a sink or source term depending on the total aerosol gradient between the FT and MBL) (Figs. 6d-f). In particular, the aerosol scavenging term is stronger in the second half of the simulation, leading to the
onset of surface precipitation in the middle of the simulation. Although precipitation continues until the end of the run, the aerosol reduction and precipitation are not strong enough to cause SCT or cloud breakup, and $Z_{inv}$ grows consistently until 6 hours before the end of the simulation.

### 4.1.3    Impact of perturbed aerosols

The very low initial $<N_a>$ (e.g., less than 30 mg$^{-1}$) in the ctrl run leads to early precipitation onset, which drives a rapid drop in $N_a$ and $N_d$ (Figs. 6a-b) and the occurrence of SCT (Fig. 5a) within the first 12 hours of simulation. [As defined by Erfani et al. (2022), an SCT occurs when LCC first falls below 50% and remains below this threshold for at least 24 hours or until the end of the simulation, whichever is shorter. This definition is designed to exclude
LCC changes attributable solely to the diurnal cycle]. This "precipitation-driven" type of SCT occurs quickly (e.g., less than 12 hours) in LES experiments with a prognostic aerosol scheme due to the positive precipitation-aerosol feedback (Yamaguchi et al., 2017; Erfani et al., 2022). Consistent with the persistent precipitation, $r_e$ remains greater than 15 μm (Fig. S3b). The aerosol scavenging sink term initially strengthens, leading to an extremely low $N_d$ (less
than 10 mg$^{-1}$), characteristic of vertically-thin horizontally-extensive layers below the $Z_{inv}$, called ultra-clean layers (UCLs) (Wood et al., 2018). Note that this feature is not seen in the CERES observations. The time-height plots of $N_d$ (figures not shown) indicate the simulation of UCLs near the inversion for the majority of ctrl run time, but only for the last 8 hours of $N_a$×3 run, because ctrl run is initialized with a value of MERRA2 $N_a$ that is 3 times smaller
than CERES $N_d$.



As the simulation progresses further, the scavenging term gradually weakens. Combined with surface fluxes of aerosols and an entrainment source term for $N_a$ (Figs. 6d-f), $N_a$ and $N_d$ are enhanced after the second night. Following the SCT, both LCC and LWP remain lower than 50% and 50 g m$^{-2}$, respectively, causing low $\tau_c$ and a weak SW CRE (Figs. S3a and 6c).

In addition, OLR is stronger in this run due to the greater longwave emission from warmer and shallower cloud tops (as seen in $Z_{inv}$) and from a warmer surface, since LCC is below 40% most of the time. MBL deepening is suppressed in this clean and precipitating environment. Two mechanisms have been proposed to explain this: 1) low aerosol concentrations correspond to weak turbulence, a decoupled MBL, and reduced entrainment (Sandu et al.,

2008); and 2) precipitation depletes LWP from the inversion layer, resulting in decreased entrainment (Blossey et al., 2013).

Although the ctrl and $N_a\times3$ runs are very distinct, the $N_a\times3$ and $N_a\times9$ runs have quite similar cloud properties in the first half of simulations. The $N_a\times9$ entrainment rate is stronger than that of $N_a\times3$ during the first and second nights likely due to enhanced entrainment

associated with $N_d$ increases in non-precipitating clouds (e.g., Igel, 2024) (See Figs. 5e, 6b). While this leads to somewhat smaller LWP during the first day of the simulation than in $N_a\times3$, the larger $N_d$ in the $N_a\times9$ run leads to the delayed initialtion of aerosol scavenging and precipitation (Figs. 6b, 6e). The Twomey effect (Platnick and Twomey, 1994) is visible in comparisons of SW CRE (Fig. 6c), where the increased $N_d$ in the $N_a\times9$ run leads to a stronger

SW CRE (Fig. 6c) than $N_a\times3$ during the first daytime despite the $N_a\times9$ run having similar LCC (Fig. 5a) and smaller LWP (Fig. 5d). Later, precipitation plays a more important role in modulating aerosol impacts: compared to the $N_a\times3$ run, the delayed onset of precipitation onset in the $N_a\times9$ run is followed by more MBL deepening (Fig. 5c), and larger LWP, $\tau_c$, and SW CRE differences between the two runs in the second half of the simulation. LCC also never

drops below 90% in this high-$N_a$ run.

Unlike the two other runs, entrainment from the FT drives the decreases in $N_a$ on average in the $N_a\times9$ run (Fig. 6d), because the MBL $N_a$ is so high that it reverses the MBL-FT aerosol gradient. Additionally, the scavenging term is a stronger sink toward the end of this run (Fig. 6e), driving a decrease in $N_a$ that leads to a sudden enhancement of precipitation.





The cloud morphology, shown in 2D maps of LWP for the $N_a \times 3$ run (Figs. 7m-p),
demonstrates the development of the mesoscale organization in the form of closed cells
within the Sc clouds early on, with the mesoscale cell size and cloud LWP increasing with
time. The Probability Distribution Functions (PDFs) of LWP and $N_d$ (Figs. 7a-h) show that
precipitation is more frequent in regions of high LWP and low-to-moderate $N_d$.




Figure 7. Probability distribution functions of LWP (a–d) and <$N_d$> (e-h) at four instantaneous times for three
LES simulations (ctrl, $N_a \times 3$, $N_a \times 9$) along the GPCI S10 (2018-07-31) trajectory. The markers present
precipitation in bins of the variable on the x-axis, and the numbers within the box in each panel show the mean
value of the variable given on the x-axis for that specific time (from top to bottom for ctrl, $N_a \times 3$, and $N_a \times 9$,
respectively). (i–t) Cloud morphology showing LWP at four times for three LES runs.

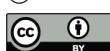



The morphology and PDF plots in Fig. 7 also demonstrate the impact of the initial aerosol concentration on cloud field development. Due to the clean environment and early SCT in the ctrl run, scattered Cu clouds form 12 hours after the initial time and are maintained throughout the simulation (Figs. 7i-l) with not much change in the cloud LWP and $N_d$ frequency distributions and the precipitation within the Cu cores (Figs. 7a-h). The cloud morphology for the $N_a$×9 run is similar to that for $N_a$×3 run 12 hours after the initial time, but the $N_a$×9 mesoscale cell size and LWP increase more rapidly. The PDFs illustrate a broader LWP spectrum with a higher probability of larger LWP in the $N_a$×9 run compared with the $N_a$×3 run 24 hours after the initial time until the end of the run (Figs. 7b-d). This spectrum broadening is associated with the precipitation onset in larger LWP bins and faster aerosol removal from the MBL (Figs. 7e-7h), which ultimately leads to more intense precipitation toward the end of the $N_a$×9 run compared to the $N_a$×3 run (Fig. 5b). Also, the $r_e$ value during the precipitation onset for the $N_a$×9 run is approximately 9 μm, which is smaller than for the $N_a$×3 (~12 μm) and the ctrl run (~15 μm) (Fig. S3b). As explained by Wood et al., (2009), precipitation can initiate with a smaller $r_e$ for larger $N_d$ and LWP.

### 4.2    Second case: Trajectory Sandu2010 (2018-07-04)

### 4.2.1    Observed characteristics

For our second case, in addition to the permanent subtropical high over the  NEP, a tropical cyclone developed to the east, visible in the surface wind pattern (Fig. 1c) and confirmed by satellite imagery and a humid FT. This is seen in the time-height plot of $q_t$ along the trajectory (figures not shown), which is located between the southeast edge of the high and the northwest edge of the cyclone. The phase space analysis (Fig. 4a) shows that this case has the highest PC2 values and one of the lowest PC1 values among the 54 selected cases. Based on along-trajectory averages of physical variables (Fig. 4b), this case is characterized by very low $P_{MSL}$, very weak stability (lowest EIS), and high 700-hPa $q$.

CERES LCC (Fig. 8a) shows a brief cloud breakup at the end of the first day and a major breakup starting in the early morning of the second day that lasts until the third night, with cloud cover restoring a few hours before the end of the run. According to the previously-





mentioned SCT definition, this does not qualify as SCT; however, others (e.g., Sandu and Stevens, 2011; Baró Pérez et al., 2024) simply define SCT as the first time LCC falls below 50%. In addition, this cloud breakup initiates during the dark hours, suggesting it might not be due to the diurnal cycle. Nevertheless, we use the term "cloud breakup" for this case throughout this study. ERA5 LCC appears out of phase with that from CERES, showing cloud

breakup at the end of the first day and cloud restoration on the morning of the second day.

AMSR observations indicate that precipitation is extremely weak in this case, with accumulated precipitation of approximately 0.1 mm after two days. Precipitation onset occurs on the second night before the major cloud breakup; however, CERES $r_e$ (Fig. S4b) does not show a significant day-to-day change and remains below 15 µm, fluctuating

between 10 µm and 13 µm in the middle of the days. The microwave (SSMI and AMSR) LWP

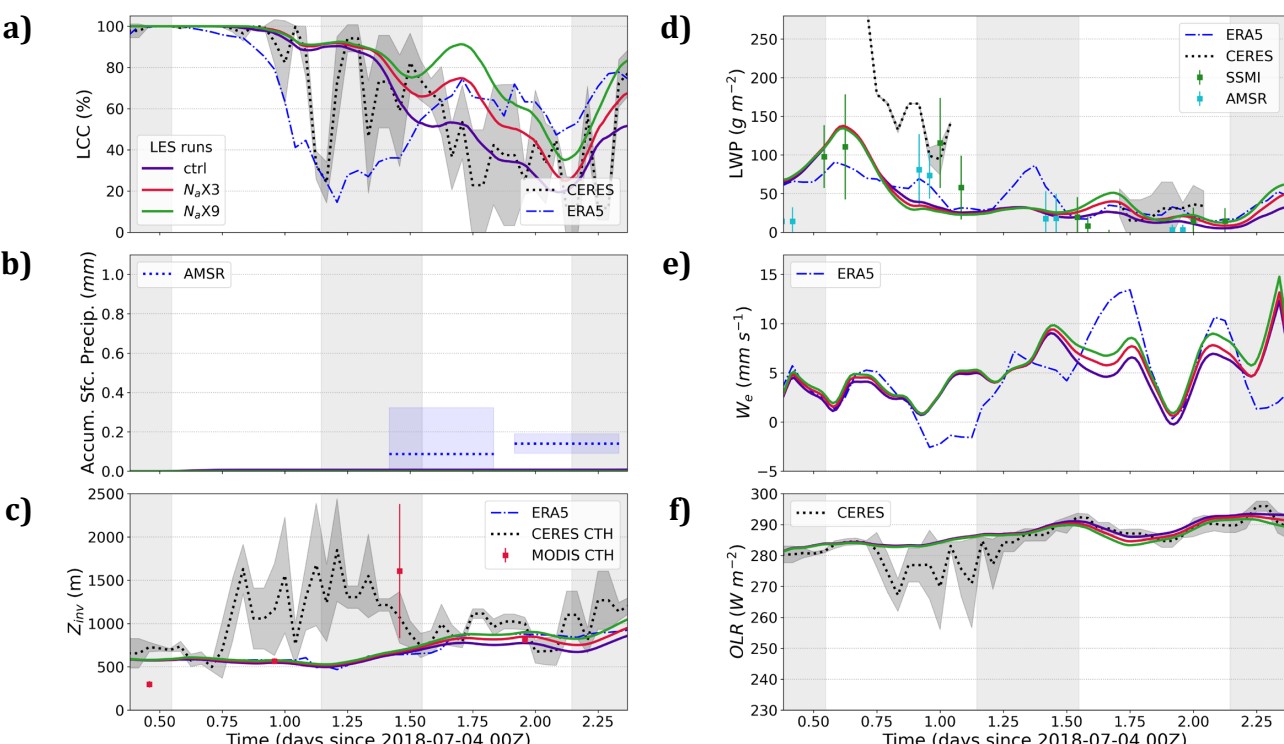

Figure 8. As in Fig. (5), but for the Sandu 2010 (2018-07-04) trajectory.





increases during the first six hours of the simulation and then decreases until the second
day (Fig. 8d). The LWP from all three observational products falls below 50 g m$^{-2}$ during the

cloud breakup. Similar to the first case, CERES LWP is generally greater than the microwave
LWP, but the two microwave retrievals, from AMSR and SSMI, remain similar.CERES CTH
and, to some extent, MODIS CTH show an unusual deepening from the middle of the first day
until the end of the following night (Fig. 8c). This seems to be due to the presence of upper-
level ice clouds, also evident in the low values of OLR during this period. Otherwise, CERES

and MODIS CTH and ERA5 $Z_{inv}$ generally agree, showing an enhancement of 200-500 m from
the start to the end of the trajectory. Based on ERA5 $Z_{inv}$, this increase begins on the second
night before the cloud breakup but slows down on the second day. This is associated with
the decoupling of the MBL, evident from the time-height plot of ERA5 $q_t$, where the $q_t$
difference between the lower and upper MBL increases along the trajectory (figure not

shown).

The initial value of the MERRA2 $<N_a>$ for this trajectory is 70 mg$^{-1}$, then $<N_a>$ gradually
decreases starting on the second night to around 30 mg$^{-1}$ by the end of the trajectory (Fig.
9a). Consistent with this, CERES $N_d$ decreases from the first day to the second day (Fig. 9b).
However, as in our other trajectory, MERRA2 $<N_a>$ around solar noon is significantly smaller

(here, by a factor of two) than the CERES $N_d$, which is clearly not physically consistent. Again
this suggests that simulations initialized with MERRA2 $<N_a>$ may struggle to match CERES
$N_d$, so other options for aerosol initialization in the MBL should be considered. Note that for
this trajectory both $<N_a>$ and $<N_d>$ values are significantly higher than the 10 mg$^{-1}$ threshold
used by Wood et al. (2018) to identify a UCL.

CERES $\tau_c$ decreases significantly over the trajectory, from an average of 15 on the first day to
an average of 3 on the second day (Fig. S4a), consistent with the reduction in both $N_d$ and
LWP. Additionally, the SW CRE from CERES demonstrates an approximately 5-fold decrease
from the first to the second day (Fig. 9c) due to cloud breakup and $\tau_c$ reduction.

### 4.2.2  Reference run (ctrl)

For this case, the ctrl run serves as our reference simulation since it performs better at
simulating most variables, such as LCC, $N_a$, $N_d$, and SW CRE, compared to the perturbed runs



(Figs. 8 and 9). This improved performance relative to the first case study likely results from the thinner clouds with smaller LWP in this case study, which do not precipitate significantly, even when initialized with the MERRA $N_a$ that are biased low relative to CERES $N_d$ (Figs. 9a-

b), but not by as much as for our first case. The noisy patterns seen in most observed variables are absent in the ctrl run, likely due to slow FT nudging. While the ctrl run does not capture the first, brief observed cloud breakup, it accurately simulates the timing and rate of the second cloud breakup on day two, and the cloud restoration on the last night (Fig. 8a). The ctrl run successfully simulates the microwave LWP pattern, showing enhancement

during the first 6 hours followed by a reduction. However, the timing of LWP reduction is earlier than microwave observations by a few hours (Fig. 8d). Despite this, the ctrl run LWP remains within the lowest bound of the microwave LWP or very close to it during this period

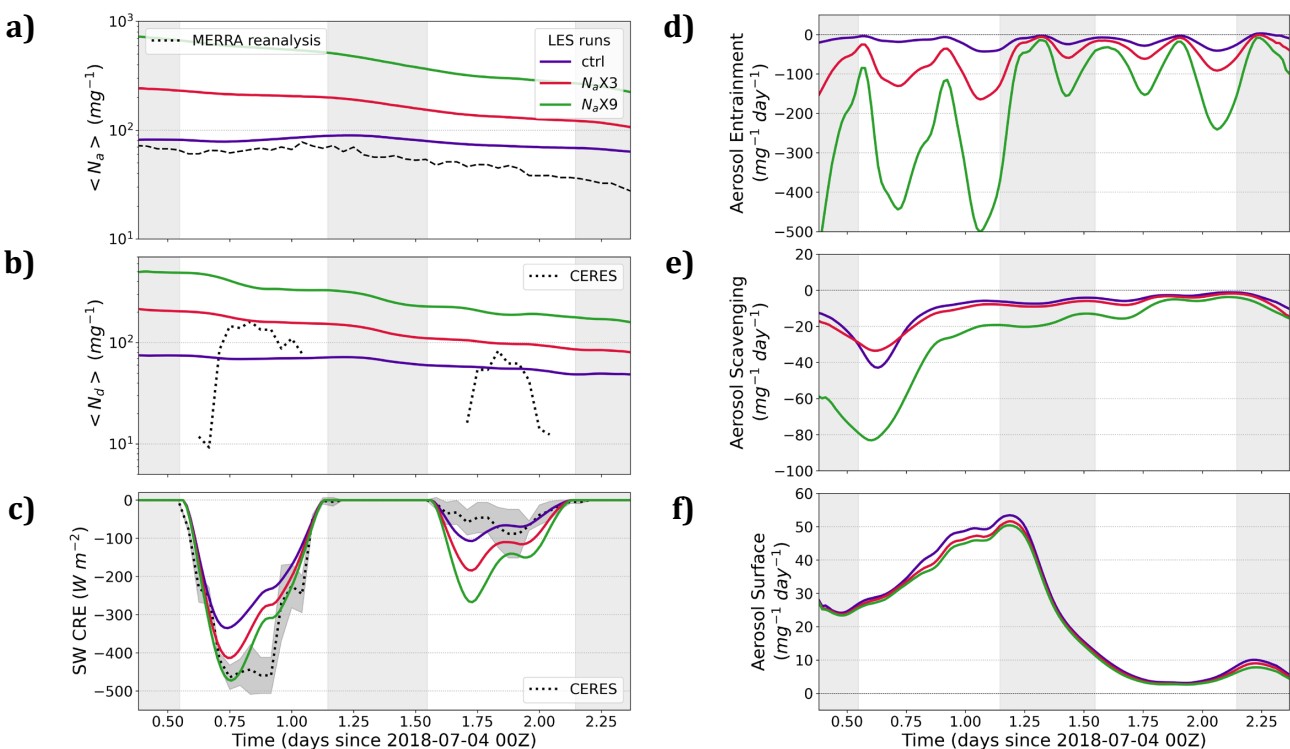

Figure 9. As in Fig. (6), but for the Sandu 2010 (2018-07-04) trajectory.





and is mostly close to the mean microwave LWP at other times. Unlike the AMSR retrieved precipitation, the ctrl run simulates no precipitation, indicating that the cloud breakup in the model is not precipitation-driven. Instead, it appears to be affected by the MBL deepening (Fig. 8c) which enhances $w_e$ during the second night (Fig. 8e). This is indicative of the deepening-warming cloud breakup mechanism, driven by the deepening and decoupling of

the MBL (Bretherton and Wyant, 1997; Wyant et al., 1997) and enhancement of entrainment near the inversion (Ackerman et al., 2004). This type of cloud breakup occurs at a much slower rate compared to precipitation-driven cloud breakup, as shown by the ctrl run for the first case (Fig. 5) and by precipitating runs in Erfani et al. (2022).

A precipitation-driven SCT is very unlikely for this case, since an environment with $N_d$

higher than 30 mg$^{-1}$ and LWP lower than 30 g m$^{-2}$ is associated with precipitation of less than 0.03 mm day$^{-1}$ (Fig. A3 in Wood et al., 2009). The increase in $w_e$ in the ctrl run before the cloud breakup during the second night is consistent with that seen in the ERA5 $w_e$, though ERA5 shows a stronger increase in $w_e$ (Fig 8e). This period is followed by a reduction in $w_e$ that lasts until the middle of the second day. The MBL deepening on the second night is not

seen in the CERES retrievals due to the presence of ice clouds during the first day. When ice clouds are absent, the ctrl run $Z_{inv}$ and OLR agree well with that from the CERES retrievals (Figs. 8c and 8f).

The ctrl run underestimates the magnitude of the observed SW CRE on the first day (Fig. 9c), despite having overcast conditions in both the ctrl run and the CERES retrievals. This

appears to be due to an underestimation of $\tau_c$ during this time (Fig. S4a), which is the result of low LWP and $N_d$ in the ctrl run (Fig. 9a), which are lower than the CERES values during the first day. The ctrl run agrees well with CERES values of $\tau_c$ and SW CRE during the second day.

Although the ctrl run simulates the general trend in MERRA2 $N_a$ and CERES $N_d$ (e.g., an

overall reduction in $<N_a>$ and $<N_d>$ from the start to the end of the run, particularly in the second half of the simulation), the rate of $<N_a>$ reduction is slower than that in MERRA2 (Fig. 9a,b). Specifically, the ctrl run $<N_a>$ decreases from 80 mg$^{-1}$ to 60 mg$^{-1}$ (Fig. 9a) and the ctrl run $<N_d>$ decreases from 70 mg$^{-1}$ to 50 mg$^{-1}$ (Fig. 9b). Based on the time-height plots of $N_a$





and $N_d$ (not shown), their values tend to be lower near the inversion but do not drop below

20 mg$^{-1}$, and therefore, these reductions are insufficient to develop UCLs. It seems that the $<N_a>$ is relaxing towards the FT values, which are below 50 mg$^{-1}$ near the inversion. During the first half of the simulation, both the aerosol scavenging sink term and the aerosol surface flux source term are stronger, while the aerosol entrainment term remains a sink term throughout the run (Figs. 9d-f). The balance among these three terms results in negligible

changes in $<N_a>$ and $<N_d>$ during the first half of the simulation, followed by a gradual reduction in the second half. Overall, the moderate initial aerosol concentrations and their slow reduction prevent the formation of large cloud droplets (as indicated by $r_e$, which remains within 8-12 µm throughout the run and is slightly smaller than CERES values; Fig. S4b) and the initiation of precipitation (Fig. 8b).

**4.2.3   Impact of perturbed aerosols**

Compared to the ctrl run, the $N_a \times 3$ run's higher $<N_a>$ and $<N_d>$ during the initial hours (Figs. 9a-b) lead to a stronger FT-MBL aerosol gradient. This results in a stronger aerosol entrainment sink term (Fig. 9d), causing a more pronounced decrease in both $<N_a>$ and $<N_d>$ over the trajectory duration. Despite this, $<N_a>$ and $<N_d>$ in the $N_a \times 3$ run remain at least

twice those in the ctrl run. Cloud breakup in the $N_a \times 3$ run occurs a few hours later than in the ctrl run, with LCC in the $N_a \times 3$ run remaining 15-20% higher than in the ctrl run during the second day. This delayed cloud breakup corresponds to a slightly stronger deepening of the MBL (approximately 100 m; Fig. 8c) and consequently, a slightly stronger $w_e$ (Fig. 8e) and lower OLR (Fig. 8f) compared to the ctrl. These results align with Sandu et al. (2008),

which demonstrated that enhanced entrainment in cases with high $N_a$ leads to stronger turbulence and MBL deepening; however, this impact is modest for this second case study.

During the first 12 hours of simulation, the higher $\tau_c$ in the $N_a \times 3$ run, compared to the ctrl run (Fig. S4a), can be attributed to the elevated initial $<N_d>$, since the initial LWP is very similar between the two runs. This corresponds with a reduction in $r_e$ by 2-3 µm in the $N_a \times 3$

run, compared to the ctrl run, primarily due to the Twomey effect during the first day, given that LCC and LWP remain relatively unchanged between the runs. Also, the change in $\tau_c$ explains the stronger SW CRE in the $N_a \times 3$ run during the first day. On the second day,





however, the stronger SW CRE in the $N_a \times 3$ run is more influenced by the higher LCC in this run (Fig. 8a) and less by $\tau_c$. In other words, the role of CF adjustment in SW CRE differences
between the runs becomes important on the second day, as evidenced by the differences in LCC.

The run with very high $N_a$, $N_a \times 9$, simulates strong aerosol entrainment and scavenging sink terms (Figs. 9d-e), leading to a faster reduction in $N_a$ and $N_d$, compared to the $N_a \times 3$ run (Figs. 9a-b). The cloud breakup in the $N_a \times 9$ run is delayed by a few hours (Fig. 8a), associated
with slightly greater MBL deepening, enhanced entrainment, and reduced OLR, compared to $N_a \times 3$ run (Figs. 8c,e,f). The higher aerosol concentration in the $N_a \times 9$ run leads to smaller $r_e$ (Fig. S4b), higher $\tau_c$ (Fig. S4a), and an increased magnitude of SW CRE (Fig. 9c). The change in SW CRE from the ctrl to the $N_a \times 3$ run is stronger than from the $N_a \times 3$ to the $N_a \times 9$ run during the first day. Considering the similar overcast conditions and LWP values across all
three runs, this highlights the dominance of the Twomey effect and albedo susceptibility. This impact diminishes on the second day as CF adjustment becomes more significant.

Maps of LWP for the ctrl run (Figs. S5i-l) show the formation of overcast Sc clouds and mesoscale organization 12 hours after the initial time, which then develop into closed cells later on. The dissipation of Sc clouds and cloud breakup are demonstrated by scattered Cu
clouds 36 hours after the initial time, but their frequency and size increase towards the end of the simulation due to cloud restoration. The PDFs of LWP (Figs. S5a-d) indicate that although the mean and median LWP values show a general decrease over time until near the end of the runs, the LWP spectrum broadens with a higher probability of larger LWP, suggesting enhanced LWP in the cores of mesoscale cells over time. The evolution of cloud
morphology and LWP PDF for the $N_a \times 3$ and $N_a \times 9$ runs is similar to that of the ctrl run, but higher MBL aerosols lead to larger mesoscale cell sizes (Figs. S5o,p,s,t) with more water in their cores, as evident from the broadening of the LWP spectrum toward the larger values (Figs. S5c-d).

For each run and at each time, the average length scale of mesoscale cells is quantified in Fig.
S6 as the wavelength below which 2/3 of the LWP variance is contained following the methodology of de Roode et al. (2004) (See their Fig. 2). During the overcast Sc regime on





day 0.5, the LWP PDF (Fig. S5a) shifts towards smaller values with increased $N_a$, as expected from the sedimentation-entrainment feedback (Ackerman et al., 2004), and domain-averaged LWP decreases with increasing $N_a$, while the length scale of mesoscale cells is larger in $N_a \times 3$ and $N_a \times 9$ runs than in ctrl run (Fig. S6a). Later on (days 1.5 and 2.0) when cloud breakup occurs and Cu clouds emerge, both mean LWP and cell size values are larger in $N_a \times 3$ and $N_a \times 9$ runs than in ctrl run. Therefore, the reduced LWP due to aerosol perturbations in the non-precipitating boundary layer at the beginning of the runs appears to be a short-lived effect in our study, since the opposite occurs when Cu under Sc becomes dominant. The influence of $N_a$ on mesoscale cell size is not well understood. Zhou and Feingold (2023) highlighted the relationship between cell size and $N_d$, but their study focused on how cell size regulates $N_d$ and LWP, rather than aerosol impact on the cell size. Further research is required to investigate the mechanisms behind the dependence of cell size on $N_a$.

Turbulence is slightly stronger in $N_a \times 9$ run than that in the other two runs before the cloud breakup and at the very end of the run. (figure not shown). Stronger turbulence might help bring more moisture to the cloud layer, hence higher LWP in the Cu cores in $N_a \times 9$ run. Note that this case has extremely weak precipitation, with precipitation in all three runs (Figs. S5a-h) being two or more orders of magnitude smaller than for the first case. Therefore, precipitation impact on turbulence is negligible.

## 5   Summary

The objective of this study is to develop an approach for selecting and analyzing a representative set of cases for studying LES model performance and how ACI and MCB affect key cloud properties in the absence of in situ observations. Utilizing ECMWF ERA5 wind data, we generate 2208 Lagrangian isobaric (950 hPa) MBL forward trajectories initialized at six locations within the subtropical NEP during JJA 2018-2021. Eliminating trajectories that pass near or over land or that include ice clouds reduces this to 1663 trajectories. Note that we retain cases with limited amounts of ice cloud to avoid selection bias for the low cloud



cases. Meteorological, cloud, aerosol, and radiation variables from reanalysis and satellite data are compiled along these trajectories to create a library of Lagrangian observations. We then use a selected number of CCFs (e.g., along-trajectory means, and differences between the beginning and end of each trajectory for WS, $q$, $\omega$, and EIS) and conduct PCA to reduce the data dimensionality. Based on the PCA results, we find that two PCs capture 43% of the variability in the CCFs. To span the meteorological diversity of the dataset, 9 trajectories are selected for each of the six initial locations in our study region, where the 9 trajectories correspond to the values of (-1.5σ, 0, 1.5σ) in the PC1-PC2 plane. This reduces the total of 1663 trajectories to a subset of 54 trajectories that span most of the variation in the CCFs, aerosol concentrations, and cloud properties relevant to their evolving radiative effect.

Some previous studies have employed aircraft measurements from intensive observational field campaigns to initialize and force Lagrangian LES runs (Blossey et al., 2021; Erfani et al., 2022). Since in-situ measurements are rare over the remote oceans, here we develop a methodology for doing routine LES modeling that is initialized with and tested against satellite retrievals and reanalysis data. In addition to meteorological data, the LES is forced with an accumulation-mode aerosol $N_a$ calculated from the MERRA-2 masses of aerosol species and their assumed particle size distributions, applying the technique described in Erfani et al. (2022) to convert aerosol mass to number concentrations. In addition, a thermodynamic "profile sharpening" method is developed to modify the initial $T$ and $q_t$ vertical profiles from ERA5 in an approach that results in cloud LWP matching that from the microwave-instrument satellite retrievals. This method leads to the instantaneous formation of a well-mixed stratiform-topped MBL in the LES.

The LES used in this study is SAM (Khairoutdinov and Randall, 2003) coupled with a prognostic aerosol scheme (Berner et al., 2013), that accounts for aerosol budget tendencies such as coalescence and interstitial scavenging, surface sources, and entrainment from the FT. From 54 Lagrangian cases, two cases are selected as examples to conduct 2-day high-resolution, large-domain Lagrangian LES experiments in order to simulate cloud evolution under observed as well as perturbed aerosol conditions. The results of a few runs for the two cases reveal that our LES is capable of simulating observed conditions when initialized with





realistic aerosol and meteorological conditions. The first case is precipitating, which implies a potential for a precipitation-driven cloud breakup if the environment is clean. Enhancing the initial aerosol concentration among different runs increases $N_d$, reduces $r_e$, enhances cloud albedo, suppresses precipitation, and increases TOA SW CRE, in agreement with previous studies (Sandu and Stevens, 2011; Yamaguchi et al., 2017; Christensen et al., 2020;

Blossey et al., 2021). $d(\mathrm{SW\,CRE})/d(N_d)$ is nonlinear, with a larger magnitude increase (more cooling) from the ctrl run to the $N_a{\times}3$ run than from the $N_a{\times}3$ run to the $N_a{\times}9$ run. This seems to be due to the positive precipitation-aerosol feedback for the ctrl run, which quickly dissipates the clouds.

The second case is non-precipitating, and the classic deepening-warming cloud breakup

happens in both the control and increased-aerosol runs. More MBL aerosol leads to stronger entrainment, more delayed cloud breakup, and a stronger SW CRE. This type of SCT was simulated in previous studies (Baró Pérez et al., 2024; Diamond et al., 2022) and seems to be more common in a polluted environment. Compared to the first case, cloud breakup occurs at a slower rate, and perturbed aerosols among different runs have a smaller impact on SW

CRE and cloud breakup due to the absence of a precipitation-aerosol feedback.

## 6    Conclusions

The PCA approach demonstrated in this study has been particularly effective in identifying a subset of Lagrangian trajectories that not only represent the variability within the PC space

but also span the full range of key cloud properties. This highlights the potential of PCA for refining complex datasets while preserving critical physical characteristics relevant to ACI and MCB studies.

An important challenge is defining a ground truth against which models could be validated, due to the considerable variability observed in cloud property datasets from reanalysis and

satellite retrieval products. This variability underscores the complexities and uncertainties inherent in both products which might affect confidence in the results. In addition, the



relatively coarse spatial resolution of these products, compared to LES resolution, could undermine the representation of diverse aerosol and cloud properties.

In general, satellite retrievals are more reliable than reanalysis products for Sc clouds. Based on climatological averages for the NEP region, ERA5 LCC is biased low when compared to MODIS (Wu et al., 2023), and since CERES LCC is based on MODIS retrievals, CERES is more robust than ERA5 when studying LCC. Microwave (AMSR and SSMI) retrievals of LWP are reliable for Sc clouds since they compare well with in-situ measurements (Painemal et al., 2016). The $Z_{inv}$ calculated in our study based on vertical profiles of ERA5 $T$ and $q_t$ appear to be more robust than other products. The CTH from MODIS retrievals is created based on data stratified within bins of $T$ each having a range of 5 °C and as such, it might not be accurate for individual cases, but it performs well on average (Eastman et al., 2017).

Over the NEP and for higher $N_a$, MERRA2 $N_a$ is biased low when compared to in-situ measurements (Erfani et al., 2022). In the future, a critical step in forcing and initializing our LES with MBL aerosols based on the values of CERES $N_d$ rather than MERRA-2 $N_a$. Given that MERRA2 $N_a$ is simulated by assimilating MODIS aerosol optical depth (which represents the optical property of aerosols throughout the column of troposphere), it can be inaccurate at certain levels and locations. CERES, on other hand, provides satellite estimates of $N_d$ in the cloud layer , which seem more reliable for Sc clouds and is consistent with other CERES products, such as TOA radiative fluxes which are considered the most accurate measurements (Su et al., 2015).

The simulations in this study demonstrate that reanalysis meteorological and aerosol data can be used for initializing and bounding LES runs, to produce realistic baseline simulations of low marine cloud fields in the absence of aircraft field campaigns. In the future, we will conduct LES experiments for a large number of Lagrangian cases from PCA results. This will enable us to synthesize valuable statistics to assess how well LES can simulate the cloud lifecycle under the "best estimate" environmental conditions, and how sensitive the simulated clouds are to variations in these driving fields. This procedure will contribute to advancing our understanding of intentional MCB efficacy under a range of representative conditions.



**Appendix A: Sharpening procedure of thermodynamic profiles**

This procedure utilizes satellite microwave retrievals of LWP to sharpen reanalysis (in particular ERA5) temperature and moisture vertical profiles through an optimization technique at a specific time near the inversion level.

**A1. Preparing variables**

We use a number of reanalysis and satellite variables to sharpen the ERA5 temperature and moisture profiles near the inversion level. At each time (generally, the time corresponding to when we initialize an LES run), the vertical profiles of ERA5 normalized liquid-water static energy ($T_l$) and total water mixing ratio ($q_t$) are calculated as:

$$T_l = T + g\frac{z}{C_p} - q_l\frac{L_v}{C_p} \qquad (A1)$$

$$q_t = q_v + q_l \qquad (A2)$$

where $T$ is temperature, $q_v$ is water vapor mixing ratio, $q_l$ is cloud liquid water mixing ratio, $z$ is height, $g$ is Earth's gravitational acceleration, $C_p$ is the specific heat of dry air at constant pressure, and $L_v$ is the latent heat of vaporization. We conduct separate calculations for the

MBL and lower FT, but first, we need to calculate the inversion height ($Z_{inv}$), which is defined as the height where $\left(\frac{d\theta_l}{dz}\right)\left(\frac{dRH}{dz}\right)$ is minimized over the atmospheric column at each time and location (Blossey et al., 2021; Erfani et al., 2022). RH is relative humidity and $\theta_l$ is liquid-water potential temperature.

First, the lower FT profile sharpening method is explained. We assume that $L_{FT}$ be a height

above the inversion where the ERA base profiles feel no impact from the MBL. This is selected to be 500 m. Therefore, for $z > Z_{inv} + L_{FT}$:

$$T_{l_{shrp}} = T_{l_{base}} \qquad (A3)$$

$$q_{t_{shrp}} = q_{t_{base}} \qquad (A4)$$





where the subscript "shrp" refers to sharpened profiles and the subscript "base" to the
baseline profiles. For $Z_{inv} < z < Z_{inv} + L_{FT}$, a line is fitted to the $T_{l_{base}}$ and $q_{t_{base}}$ profiles away
from the inversion (e.g., $Z_{inv} + L_{FT} < z < Z_{inv} + 3L_{FT}$) and is extrapolated down to the inversion.

Now, the MBL profile sharpening method is described. At the top of the MBL ($z = Z_{inv}$), the
values are calculated as:

$$T_{l_{shrp}} = T_{l_{base}} + \Delta T_{l_{inv}} \tag{A5}$$

$$q_{t_{shrp}} = q_{t_{base}} - \Delta q_{t_{inv}} \tag{A6}$$

where $\Delta q_{t_{inv}}$ and $\Delta T_{l_{inv}}$ are the differences in $q_t$ and $T_l$ between the lower FT and upper
MBL. The initial values are provided to the code, and the optimization function finds the
adjusted values. The profiles within the MBL ($z < Z_{inv}$) are calculated as:

$$T_{l_{shrp}} = \min\left(T_{l_{inv}}, T_{l_{base}}\right) \tag{A7}$$

$$q_{t_{shrp}} = \max\left(q_{t_{inv}}, q_{t_{base}}\right) \tag{A8}$$

Finally, we utilize the density temperature as:

$$T_\rho = T(1 + 0.61q_v - q_l) \tag{A9}$$

## A2. Optimization

An optimization algorithm is created that takes reanalysis $T_{l_{base}}$, $q_{t_{base}}$, $\Delta T_{l_{inv}}$, $\Delta q_{t_{inv}}$, $Z_{inv}$,
and microwave LWP as inputs and computes $T_{l_{shrp}}$ and $q_{t_{shrp}}$ as described in Appendix A1.
It then calculates the LWP of the baseline and sharpened profiles by vertical integration of $q_l$.
Note that "saturation adjustment" must be employed to calculate $q_l$ at each height.
Saturation adjustment is a common practice in weather and climate modeling of clouds, and
it means that any vapor in excess of saturation is converted to condensate (McDonald, 1963).
Thereafter, a cost function, $A$, is calculated in order to quantify how well the resulting





sharpened profile matches the microwave LWP while preserving the vertical integrals of the ERA5 $T_\rho$ and $q_t$ profiles:

$$A = f_1\left(LWP_{microwave} - LWP_{shrp}\right)^2 +$$

$$f_2\left(\int_0^h T_{\rho_{base}}\rho dz - \int_0^h T_{\rho_{shrp}}\rho dz\right)^2 + f_3\left(TWP_{base} - TWP_{shrp}\right)^2 \tag{A10}$$

where TWP is the total water path, calculated by integrating $q_t$ from surface to an arbitrary height, $h$. Here, a value of 3000 m is sufficient for the profile sharpening of marine Sc clouds. Parameters $f_1$, $f_2$, and $f_3$ are selected in a way to keep the values of three terms on the right-hand side in the same order of magnitude: $f_1 = \frac{1}{(0.01\ kg\ m^{-2})^2}$, $f_2 = f_3\left(\frac{C_p}{L_v}\right)^2$, $f_3 = \frac{1}{F^2}$, where $F$ is an input to the optimization function and its optimized values are in the range of $F$ is 10-

30 kg m$^{-2}$ for our cases. The optimization function is then prepared to minimize the variable $A$ by varying initial values of $\Delta T_{l_{inv}}$, $\Delta q_{t_{inv}}$, and $F$, but keeping the microwave LWP, $Z_{inv}$, $T_{l_{base}}$, and $q_{t_{base}}$ constant. The optimization function provides the optimum values of $\Delta T_{l_{inv}}$, $\Delta q_{t_{inv}}$, and $F$, which then will be used to calculate the ultimate $T_{l_{shrp}}$ and $q_{t_{shrp}}$ profiles (Fig. A1).


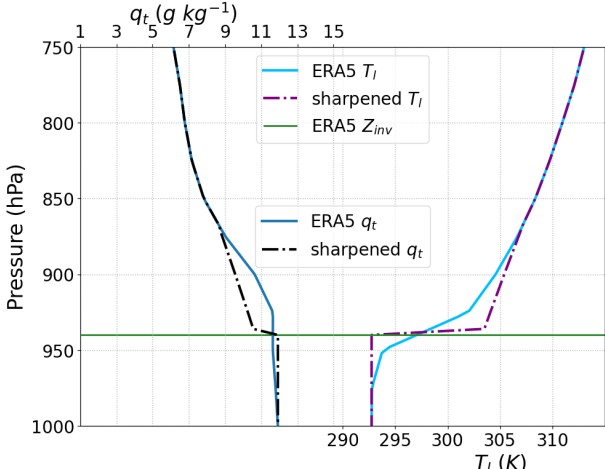

Figure A1. Vertical profiles of $q_t$ and $T_l$ from ERA5 and the sharpened versions of these profiles used to initialize the Sandu 2010 (2018-07-04) trajectory.



***Code and data availability:*** The required observational/reanalysis data, input forcing files,
LES model setup scripts, LES outputs, and Python codes to reproduce the results of this study
are provided on Zenodo: https://doi.org/10.5281/zenodo.13917317 (Erfani et al., 2024).
The "uw-trajectory" Python package for compiling reanalysis data and satellite retrievals
along the Lagrangian trajectories is available on GitHub: https://github.com/e-erfani/uw-
trajectory/ and on Zenodo: https://doi.org/10.5281/zenodo.13917362 (Erfani, 2024).
CERES SYN1deg data is available at https://ceres.larc.nasa.gov/ (NASA, 2016). AMSR and
SSMI data are obtained from www.remss.com/missions/ (Wentz et al., 2012, 2014). ERA5
data is accessible from https://doi.org/10.24381/cds.adbb2d47 (Hersbach et al., 2020).
MERRA2 data is available from https://doi.org/10.5067/VJAFPLI1CSIV (GMAO, 2015). The
SAM          code          is          publicly          accessible          at
https://you.stonybrook.edu/somas/people/faculty/marat-khairoutdinov/sam/
(Khairoutdinov, 2022).

***Author contributions:*** All co-authors contributed to the conceptualization, methodology,
and discussions about interpreting the results. RW and SD guided the project and provided
funding. EE developed the Python codes and conducted statistical analysis and LES modeling
with inputs from other co-authors. RE contributed to developing trajectories. PB contributed
to LES model development. EE drafted the manuscript and all co-authors provided edits and
revisions.

***Competing interests:*** The authors declare that no competing interests are present.

***Acknowledgments:*** This study was primarily supported by NOAA's Climate Program Office
Earth's Radiation Budget (ERB) Program, Grant NA22OAR4310474, as well as through the
University of Washington's Marine Cloud Brightening Program, which is funded by the
generous support of a growing consortium of individual and foundation donors. This





publication is also partially funded by the Cooperative Institute for Climate, Ocean, and Ecosystem Studies (CICOES) under NOAA Cooperative Agreement NA20OAR4320271, Contribution No. 2024-1410. This work conducted LES experiments on Bridges-2 at Pittsburgh Supercomputing Center through allocation EES210037 (Brown et al., 2021) from the Advanced Cyberinfrastructure Coordination Ecosystem: Services & Support (ACCESS) program, which is supported by National Science Foundation grants #2138259, #2138286, #2138307, #2137603, and #2138296 (Boerner et al., 2023). R. Eastman was supported by NASA grant 0NSSC19K1274. We appreciate discussions with Dennis Hartmann and Philip Rasch that contributed to the improvement of the final results.

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
