# Peer review of "Building a comprehensive library of observed Lagrangian trajectories for testing modeled cloud evolution, aerosol-cloud interactions, and marine cloud brightening"

_EGUsphere, 2024_

## Referee Comment (RC2)

**Review of Erfani et al. "Building a comprehensive library…"**

This work presents a novel methodology to generate representative samples of marine stratocumulus for LES simulation based on remote observations and reanalysis. The motivation, methods, and analysis are well-written and thoroughly documented. I have a few major comments related to the trajectory and PCA methods and the performance of the LES which warrant additional justification and explanation. After these minor revisions, I believe this work will make an excellent contribution to the literature.

**Major Comments**:

- Trajectory modeling (section 2.2): I would like to see further details about how the trajectory forward modeling is performed.
  - Which fields are assimilated and which, if any, are freely evolving?
  - Why does the uw-trajectory package utilize such a huge gridbox (2deg x 2deg) to provide input data when finer-resolution data is available from many resources?
  - I am also confused about the purpose of creating an 86h trajectory when only 48h trajectories are relevant to the study.
- PCA & dimensionality reduction: I would like to see better justification for the PCA and sampling methods used, specifically…
  - Can you describe the meaning and purpose of including the difference in each CCF between the beginning and end of the trajectory? What quality of the Sc are you trying to capture by distinguishing the mean and the difference?
  - I'm confused by the decision to remove SST and P_MSL from the list of standard CCFs, considering that PCA is already a dimensionality reduction. The covariance values cited in L260-261 to exclude these factors seem like an arbitrary cutoff, especially considering your determination that correlations above 0.14 are statistically significant (L275-279). Since the ultimate selection and display of factors is performed along PCA space, you could continue to include SST and/or P_MSL in the analysis while still ultimately selecting two principal components to represent the majority of variability.
  - Related to the point above, why did you stop at 2 principal components to analyze variability in the data? PC3 is not that much less important than PC2 (15% versus 19%) and is strongly correlated with delta-omega in a way that is not captured by the first two PCs.

- o Given that your eventual analysis samples from the original dataset rather than strictly along the axes of PC1 and PC2, the downsampled data includes axes of variability that are beyond what PC1 and PC2 alone represent, which questions the utility of the PCA altogether. How well do these downsampled trajectories span higher order PC spaces? If you were to randomly sample your trajectories along the CCF axes instead, would you obtain a substantially less representative downsampled set?
  - o How did you select the two trajectories used for LES modeling? In terms of both CCFs and the two PCs, they appear more similar than not within the spread of the data.
- Mismatch in performance of the CTRL case between the two LES examples: While the authors do an excellent job discussing the impacts of varying Na within a given LES setup, I would like to see further explanation or analysis of the fact that the MERRA-2 diagnosed aerosol concentration is not a reliable choice to use for initializing the model aerosol concentration.
  - o Given that this work seeks to develop methods for analyzing MCB and SCT, the ability of the model to predict cloud behavior under different aerosol concentrations is of utmost importance. Considering that the LES cannot reliably predict the real cloud behavior using a renanalysis-derived value of Na, how much would you trust the model to predict a *hypothetical* brightened or thinned cloud, where there is no observation for comparison?
  - o In L895, you propose initializing with CERES Nd instead, but figures 8 and 9 both indicate (a) very poor temporal coverage; and (b) strong variability in the CERES value. How would you mitigate these issues to come up with a trustworthy value for your forward model?
  - o If MERRA-2 offers a predicted value of Nd, it would be useful to show that against the CERES retrieved Nd and LES experiments. A close match between MERRA-2 and CERES Nd would indicate that the prognosed MERRA-2 Na is actually useful, but that some difference or deficiency in the aerosol-cloud parameterizations in the LES leads to the underprediction of Nd in certain cases.

**Minor Comments:**

- L87-88: LES resolves some subgrid-scale turbulence that is not resolved in a GCM, but it does not actually represent aerosols or cloud microphysics any differently than most GCM approaches (moment and modal models), unless you are specifically referring to using a sectional or superdroplet approach.

- L116+: It would be helpful to clarify at the start of this paragraph that you are referring to the present study or the previous Erfani 2022 work.
- Figure S1 is fuzzy / poor image quality
- L602: typo in "initiation"

---

## Author Response (AR1)

**Response to the reviewer's comments on the manuscript**

Title: Building a comprehensive library of observed Lagrangian trajectories for testing modeled cloud evolution, aerosol-cloud interactions, and marine cloud brightening
By: Ehsan Erfani, Robert Wood, Peter Blossey, Sarah J. Doherty, Ryan Eastman
Article reference: egusphere-2024-3232

We wish to thank the reviewers for their detailed and helpful comments on our paper. As you will see below, we have responded to all of the comments with revisions designed to address the concerns of the reviewers. In the following response, the original reviewer comments appear in blue, and our responses appear in black. In addition to the reviewers' comments, we noticed two minor issues and corrected them: 1) Figures 1b&c now show trajectories for a 48-hour period instead of 86 hours to be consist with Figure 1a and the rest of the paper; and 2) Figure 2 previously showed the time-height of Na for the ctrl LES runs, but it has been corrected to display MERRA2 Na instead.

**REVIEWER COMMENTS:**

**Reviewer #1:**

Review of "Building a comprehensive library of observed Lagrangian trajectories for testing modeled cloud evolution, aerosol-cloud interactions, and marine cloud brightening" by Erfani et al.

Summary:

This paper presents an approach to construct a library of Lagrangian trajectories and meteorological factors using reanalysis and satellite data, aiming to represent a comprehensive range of environmental conditions typical of low marine cloud regions for process modeling studies. PCA analysis was performed to identify representative trajectories that capture the variability and co-variability of observed cloud-controlling factors, aerosols, and cloud fields. The authors subsequently selected two cases to initiate LES simulations based purely on satellite and reanalysis data, and demonstrate that their LES model is capable of simulating observed conditions. The paper concludes that the strength of the aerosol-induced cloud radiative effect is sensitive to "precipitation-aerosol feedback".

I enjoyed reading this paper, which is well-written and well-organized. I believe it will make a valuable contribution to ACP. I have only a few minor comments, which are detailed below.

Major comments:

The accurate tracking of trajectories is a fundamental aspect of this study and the authors' upcoming work. While some details on the trajectory tracking method are

provided in Section 2.2, it would be beneficial to further discuss any potential limitations or assumptions associated with the trajectory code used in this study. Are there any known biases or uncertainties in the trajectory data that readers should be aware of? Additionally, given that HYSPLIT is one of the most widely used tools for trajectory tracking, I was wondering why the authors chose not to use this model for generating the trajectories in their study. By the way, recent study have compared trajectory models that rely on wind speed for tracking (e.g., HYSPLIT) with actual cloud movements, and have found notable discrepancies between the two (Larson et al., 2022). Therefore, It would strengthen the manuscript if the authors could include a more detailed discussion of the uncertainties associated with their chosen trajectory tracking method.

Reference:

Larson K M, Shand L, Staid A, et al. An optical flow approach to tracking ship track behavior using GOES-R satellite imagery[J]. IEEE Journal of Selected Topics in Applied Earth Observations and Remote Sensing, 2022, 15: 6272-6282.

We thank the reviewer for their constructive comments and for recognizing the contributions of our work. To answer the above comment, the trajectory tracking method we used relies on the UW trajectory generation code, which has been applied successfully in several studies of cloud evolution (e.g., Eastman and Wood, 2016; Bretherton et al., 2010). This method uses ERA5 u and v components of horizontal wind to advect air parcels isobarically and assumes minimal vertical motion compared to horizontal motion within the MBL section of stratocumulus deck regions. Bretherton et al. (2010) and Eastman et al. (2017) demonstrated that ECMWF ERA wind fields are appropriate for generating Lagrangian trajectories in the Eastern Pacific. The HYSPLIT model, although widely used, does not natively utilize ERA5 data in its raw format, and requires additional preprocessing. In contrast, our trajectory generation code directly uses ERA5 data, which ensures consistency with the "uw-trajectory" Python package utilized later in this study for data extraction. This consistency between datasets and tools simplifies our workflow.

We acknowledge the inherent uncertainties associated with trajectory methods. Larson et al. (2022) noted that trajectory models, which rely on wind fields, may not always align perfectly with observed cloud motion, possibly due to vertical wind shear. Such wind shear might result in non-negligible horizontal advective tendencies in our quasi-Lagrangian framework because the trajectory is not Lagrangian at all levels (e.g., Blossey et al., 2021). Despite these limitations, isobaric trajectories have been shown to effectively capture the dominant motions in the Sc-topped MBL (e.g., Sandu et al., 2010) and tracers such as CO and $O_3$ exhibited high coherence along such trajectories over a two-day period during the CSET field campaign (Mohrmann et al., 2019). Our choice of the UW trajectory generation code is motivated by its demonstrated effectiveness in prior research and its compatibility with the ERA5 dataset used throughout this study.

To address the reviewer's comment, we now have cited Larson et al. (2022) and expanded the description of UW trajectory generation code in Sect. 2.2 to highlight

the assumptions and limitations with the trajectory generation code (beginning at L196):

"We employ a trajectory generation code developed at the University of Washington (UW) that has been previously applied successfully in several studies of cloud evolution (Bretherton et al., 2010; Eastman and Wood, 2016). This code is used to generate a total of 2208 Lagrangian isobaric (950 hPa) forward trajectories. The resulting trajectories cover a timespan of 86 hours (3.5 days) and are all from the summer months (June, July, and August, or JJA) in the years 2018-2021. This method uses ERA5 u and v components of horizontal wind to advect air parcels isobarically and assumes minimal vertical motion compared to horizontal motion within the MBL section of stratocumulus deck regions. Bretherton et al. (2010) and Eastman et al. (2017) demonstrated that ECMWF ERA wind fields are appropriate for generating Lagrangian trajectories in the Eastern Pacific. Some widely used trajectory models do not natively utilize ERA5 data in its raw format, and require additional preprocessing. In contrast, our trajectory generation code directly uses ERA5 data, which ensures consistency with the "uw-trajectory" Python package utilized later in this study for data extraction.

We acknowledge the inherent uncertainties associated with trajectory methods. Larson et al. (2022) noted that trajectory models, which rely on wind fields, may not always align perfectly with observed cloud motion, possibly due to vertical wind shear. Such wind shear might result in non-negligible horizontal advective tendencies in our quasi-Lagrangian framework because the trajectory is not Lagrangian at all levels (e.g., Blossey et al., 2021). Despite these limitations, isobaric trajectories have been shown to effectively capture the dominant motions in the Sc-topped MBL (e.g., Sandu et al., 2010) and tracers such as CO and $O_3$ exhibited high coherence along such trajectories over a two-day period during the CSET field campaign (Mohrmann et al., 2019). Our choice of the UW trajectory generation code is motivated by its demonstrated effectiveness in prior research and its compatibility with the ERA5 dataset used throughout this study."

Specific Comments:

L69: Change "altered." to "altered" (remove the period).
Corrected.

L204: Include a brief explanation of the principle behind the trajectory generation code and its difference from 'uw-trajectory.'
Based on the major comment by the reviewer and as explained previously, we elaborate on the trajectory generation code. To clarify the difference between that and uw-trajectory, we added this sentence after explaining uw-trajectory in Sect. 2.2 (beginning at L230):
"Note that the trajectory generation code determines the parcel pathways based on ERA5 wind data, while the uw-trajectory Python package integrates this information to extract cloud and meteorological data along the trajectories."

The observational and reanalysis datasets used in this study (Table 1) have significantly coarser spatial resolutions compared to the LES simulations, which limits direct alignment between the two. For example, setting the observational/reanalysis domain size to match the LES domain (~50 km) would result in no grid or only a single grid cell for datasets with coarser spatial resolutions, which is 1° (e.g., CERES, MODIS). The averaging method described in the manuscript has been employed in several previous studies (e.g., Eastman and Wood, 2016; Mohrmann et al., 2019; Blossey et al., 2021; Erfani et al., 2022). In addition, clouds in these regions are organized on scales larger than the finest reanalysis resolution and typical LES domain sizes. Averaging over only those finer scales would cause derived forcings to disproportionately reflect localized cloud variations rather than capturing their broader organization. This would risk simulations being biased toward thicker or thinner parts of the cloud system rather than being representative of the overall cloud field and its internal variability. The 2° × 2° averaging helps mitigate this issue by ensuring that simulations are more representative of the full range of cloud structures in the region.

To address this comment and a comment by reviewer #2, we have added to the manuscript, Sect. 2.2 (beginning at L225), the following:

[revised manuscript text omitted]

L871: Consider renaming 'Conclusions' to 'Discussion' and relocating it before the 'Summary.'

Done.

**Reviewer #2:**

**Review** of Erfani et al. "Building a comprehensive library…"

**This** work presents a novel methodology to generate representative samples of marine stratocumulus for LES simulation based on remote observations and reanalysis. The motivation, methods, and analysis are well-written and thoroughly documented. I have a few major comments related to the trajectory and PCA methods and the performance of the LES which warrant additional justification and explanation. After these minor revisions, I believe this work will make an excellent contribution to the literature.

**Major** Comments:

● Trajectory modeling (section 2.2): I would like to see further details about how the trajectory forward modeling is performed.
   ○ Which fields are assimilated and which, if any, are freely evolving?

We thank the reviewer for their constructive comments and for recognizing the contributions of our work. To answer the above comment, the trajectory generation code uses ERA5 horizontal wind fields (u and v components) to advect air parcels. These fields are taken from the ERA5 reanalysis dataset, which provides hourly data at a spatial resolution of 0.25° × 0.25°. Discussing the ERA5 Data Assimilation (DA) system to prepare horizontal wind components fields is beyond the scope of this study. The trajectories themselves are freely evolving in the sense that they are calculated based on the advection of air parcels by the wind fields. However, the meteorological and cloud properties along the trajectories (e.g., temperature, humidity, cloud fraction) are extracted from reanalysis and satellite data and are not prognosed by the trajectory model.

To address this comment and a major comment by reviewer #1, we have added a more detailed explanation of the trajectory modeling process in Section 2.2 (beginning at L196):

"We employ a trajectory generation code developed at the University of Washington (UW) that has been previously applied successfully in several studies of cloud evolution (Bretherton et al., 2010; Eastman and Wood, 2016). This code is used to generate a total of 2208 Lagrangian isobaric (950 hPa) forward trajectories. The resulting trajectories cover a timespan of 86 hours (3.5 days) and are all from the summer months (June, July, and August, or JJA) in the years 2018-2021. This method uses ERA5 u and v components of horizontal wind to advect air parcels isobarically and assumes minimal vertical

motion compared to horizontal motion within the MBL section of stratocumulus deck regions. Bretherton et al. (2010) and Eastman et al. (2017) demonstrated that ECMWF ERA wind fields are appropriate for generating Lagrangian trajectories in the Eastern Pacific. Some widely used trajectory models do not natively utilize ERA5 data in its raw format, and require additional preprocessing. In contrast, our trajectory generation code directly uses ERA5 data, which ensures consistency with the "uw-trajectory" Python package utilized later in this study for data extraction.

We acknowledge the inherent uncertainties associated with trajectory methods. Larson et al. (2022) noted that trajectory models, which rely on wind fields, may not always align perfectly with observed cloud motion, possibly due to vertical wind shear. Such wind shear might result in non-negligible horizontal advective tendencies in our quasi-Lagrangian framework because the trajectory is not Lagrangian at all levels (e.g., Blossey et al., 2021). Despite these limitations, isobaric trajectories have been shown to effectively capture the dominant motions in the Sc-topped MBL (e.g., Sandu et al., 2010) and tracers such as CO and $O_3$ exhibited high coherence along such trajectories over a two-day period during the CSET field campaign (Mohrmann et al., 2019). Our choice of the UW trajectory generation code is motivated by its demonstrated effectiveness in prior research and its compatibility with the ERA5 dataset used throughout this study."

- o Why does the uw-trajectory package utilize such a huge gridbox (2deg x 2deg) to provide input data when finer-resolution data is available from many resources?

We have added this (in Sect. 2.2, beginning at L225) to explain the rationale for using a 2° × 2° box:

"The uw-trajectory package provides data averaged over a 2° × 2° box centered on each trajectory point at each time. This sample size is consistent with previous studies (e.g., Eastman and Wood, 2016; Mohrmann et al., 2019) and ensures sufficient data around each trajectory point is selected based on the spatial scales of the satellite cloud and reanalysis meteorological properties, as shown in Table 1. Previous work has shown that the results are largely insensitive to the exact spatial scale of averaging within a range of 100–400 km (Eastman and Wood, 2016). Note that the trajectory generation code determines the parcel pathways based on ERA5 wind data, while the uw-trajectory Python package integrates this information to extract cloud and meteorological data along the trajectories.

The trajectory accuracy is approximately 100 km per day based on ERA5 low-level wind uncertainties (~1 m/s), but a more conservative worst-case estimate suggests positional errors up to 170 km per day or 340 km per two days (assuming a 10% error in 20 m/s wind speeds). However, Eastman et al. (2016) showed that the e-folding length scale of cloud fraction correlations (i.e., the distance at which correlations between cloud fraction vectors decline by a factor of $1/e$) in stratocumulus decks is around 450 km. This suggests that cloud properties remain correlated within this range, meaning that even if our method does not track the same features precisely, it still samples a similar cloud scene. As clouds in these regions are organized on scales larger than the finest reanalysis resolution and typical LES domain sizes, averaging over only those finer scales would cause derived forcings to disproportionately reflect localized cloud variations rather than capturing their broader organization. This would risk simulations being biased toward thicker or thinner parts of the cloud system rather than being representative of the overall cloud field and its internal variability. The 2° × 2° averaging helps mitigate this issue by ensuring that simulations are more representative of the full range of cloud structures in the region."

o I am also confused about the purpose of creating an 86h trajectory when only 48h trajectories are relevant to the study.

To address this comment, we added this explanation to Sect. 2.2 (beginning at L249):

"Trajectories are generated for a total of 86 hours to capture the full evolution of air masses as they move through the SCT region, which may be relevant for future studies. However, only the first 48 hours of each trajectory are used for the purpose of LES modeling in this study. This is because the aerosol lifecycle within the MBL typically lasts 1-2 days (Lewis and Schwartz, 2004), and the accuracy of calculated trajectories diminishes significantly beyond 2 days due to the accumulation of errors in atmospheric analyses (Stohl and Seibert, 1998). Also, reproducing the evolution of Sc clouds over relatively short time frames is a significant challenge (e.g., Stevens et al., 2005) and worthy of study in its own right, independent of the SCT. Given the importance of precipitation to boundary layer dynamics and cloud cover (Stevens et al, 1998; Yamaguchi et al., 2017) and the uncertainty of aerosol concentrations in the boundary layer (e.g., Erfani et al., 2022, Appendix A), a careful study of the evolution of the Sc-topped MBL ahead of its transition to a trade Cu boundary layer is warranted."

- PCA & dimensionality reduction: I would like to see better justification for the PCA and sampling methods used, specifically…
  - Can you describe the meaning and purpose of including the difference in each CCF between the beginning and end of the trajectory? What quality of the Sc are you trying to capture by distinguishing the mean and the difference?

    The inclusion of the difference in each CCF between the beginning and end of the trajectory (e.g., ΔEIS, Δ$q$, Δ$\omega$, ΔWS) is intended to capture the temporal evolution of the meteorological conditions along the trajectory. This is particularly important for understanding how changes in these factors affect the MBL and Sc cloud properties. For example, a considerable decrease in EIS along a trajectory indicates enhanced MBL instability, which can drive MBL deepening, cloud breakup, and precipitation onset. By including both the mean and the difference, our goal is to capture the dynamic processes that influence cloud evolution.

    We have added the following explanation to the manuscript, Sect. 2.3 (beginning at L301):

    "The inclusion of both the mean and the difference in each CCF along the trajectory is intended to capture the dynamic processes that influence cloud evolution. In particular, the differences quantify how changes in CCFs over time affect the MBL and cloud properties along the trajectory."

  - I'm confused by the decision to remove SST and P_MSL from the list of standard CCFs, considering that PCA is already a dimensionality reduction. The covariance values cited in L260-261 to exclude these factors seem like an arbitrary cutoff, especially considering your determination that correlations above 0.14 are statistically significant (L275-279). Since the ultimate selection and display of factors is performed along PCA space, you could continue to include SST and/or P_MSL in the analysis while still ultimately selecting two principal components to represent the majority of variability.

    The decision to exclude SST and $P_{MSL}$ from the PCA was based on their high covariance with other CCFs. When two variables co-vary strongly with each other, excluding one as a PCA input helps eliminate redundancy, reduce complexity, enhance explainability, and improve PCA performance in identifying independent modes of variability. Including SST and $P_{MSL}$ would increase the number of input variables from 8 to 12, which would reduce the total variance percentage explained by the first two PCs (Fig. R1). Also, the contribution of EIS, WS, $q$ and $\omega$ to each PC would be changed, as the PCA would allocate some portions of variability to SST and $P_{MSL}$, which are not

independent of other CCFs. While PCA is a dimensionality reduction technique, it is still important to focus on a set of input variables with low co-variability among themselves to ensure meaningful and interpretable results.

We have added the following explanation to the manuscript, Sect. 2.3 (beginning at L306):

"Here we excluded SST and $P_{MSL}$ from the PCA because they have high co-variability with other CCFs, as characterized by the correlation coefficient (R-value). For example, the R between SST and EIS is -0.6, and $\Delta$SST and $\Delta P_{MSL}$ are highly correlated with $WS_{10m}$ (0.6 and -0.5, respectively; see Fig. S1). When one or two variables co-vary strongly with others, excluding them as PCA inputs helps eliminate redundancy, reduce complexity, enhance explainability, and improve PCA performance in identifying independent modes of variability. While PCA is a dimensionality reduction technique, it is still important to focus on a set of input variables with low co-variability among themselves."

o Related to the point above, why did you stop at 2 principal components to analyze variability in the data? PC3 is not that much less important than PC2 (15% versus 19%) and is strongly correlated with delta-omega in a way that is not captured by the first two PCs.

Focusing on PC1 and PC2 allows us to simplify the analysis while still capturing the most significant modes of variability. While PC3 explains an additional 14% of the variability and is strongly correlated with $\Delta\omega$ (R = -0.69) and $\omega$ (R = 0.53), variability in $\omega$ and $\Delta\omega$ is partially captured by PC1 and PC2, respectively. We perform a linear regression of $\omega$ on PC1 and PC2 to find the best linear combination of a*PC1+b*PC2. The correlation between $\omega$ and this combination is 0.48. Similarly, the correlation between $\Delta\omega$ and c*PC1+d*PC2 is 0.23. These values give the variance explained by the combination of the first two PCs for $\omega$ and $\Delta\omega$. Including PC3 in the phase space analysis would require mapping data points in a 3-dimensional space, with each PC on one axis, and selecting 27 representative points for each location, totaling 162 points for 6 locations. Such a large number of trajectories is beyond our resources for conducting high-resolution, detailed LES in future work.

We have added the following explanation to the manuscript, Sect. 3.1 (beginning at L458):

"Focusing on PC1 and PC2 allows us to simplify the analysis while still capturing the most significant modes of variability. While PC3 explains an additional 14% of the variability and is strongly correlated with $\Delta\omega$ (R = -0.69) and $\omega$ (R = 0.53), variability in $\omega$ and $\Delta\omega$ is partially captured by PC1 and PC2, respectively. We

perform a linear regression of ω on PC1 and PC2 to find the best linear combination of a*PC1+b*PC2. The correlation between ω and this combination is 0.48. Similarly, the correlation between Δω and c*PC1+d*PC2 is 0.23. These values give the variance explained by the combination of the first two PCs for ω and Δω. In addition, including PC3 in further analysis would lead to a much larger number of representative trajectories, which is beyond our resources for conducting high-resolution, detailed LES in future work."

o Given that your eventual analysis samples from the original dataset rather than strictly along the axes of PC1 and PC2, the downsampled data includes axes of variability that are beyond what PC1 and PC2 alone represent, which questions the utility of the PCA altogether. How well do these downsampled trajectories span higher order PC spaces? If you were to randomly sample your trajectories along the CCF axes instead, would you obtain a substantially less representative downsampled set?

One limitation of applying PCA and using a couple of PCs to downsample data is that it cannot represent the full variability of the dataset. However, selecting the first two PCs ensures that the highest variability is captured, as PCA is designed to identify the most significant modes of variation and use them to guide the selection of representative trajectories. To address the reviewer's comment, we conducted a sensitivity analysis by randomly sampling trajectories for each location (Fig. R2). This analysis shows that while random sampling demonstrates some ability to capture variability, it provides a less representative subset of trajectories, especially within the extreme ranges of the datasets, compared to the PCA-based selection method. This highlights the PCA capability in identifying trajectories that represent the variability in the dataset.

We have added the following explanation in Sect. 3.2 (beginning at L512):

"One limitation of our downsampling method is that it cannot represent the full variability of the dataset. We conduct a sensitivity analysis by randomly sampling trajectories for each location (figure not shown). This analysis reveals that while random sampling demonstrates some ability to capture variability, it provides a less representative subset of trajectories, especially within the extreme ranges of the datasets, compared to the PCA-based selection method. This highlights the PCA capability in identifying trajectories that represent the variability in the dataset."

 How did you select the two trajectories used for LES modeling? In terms of both CCFs and the two PCs, they appear more similar than not within the spread of the data.

We have added a few sentences in the first paragraph of Sect. 4 (beginning at L521) to clarify how the two trajectories are distinct:

"Here we take two of the 54 selected trajectories identified as covering the range in cloud variability at our six representative sites in the NEP region and use them to demonstrate our approach to testing the LES-simulated cloud evolution against observed cloud evolution starting in the Sc region and moving toward the more Cu-dominated region. The two trajectories used for LES modeling were selected to represent distinct regions of the PC1-PC2 phase space (Fig. 4a). In particular, as seen in Fig. 4b, the first case exhibits stronger surface pressure, higher LWP, higher CTH, lower FT $q$, stronger stability, and higher cloud coverage compared to the second case. In this way, and as explained throughout this section, these cases test the LES model under different combinations of CCFs and cloud properties. In a later study, this approach will be used to statistically analyze model performance across all 54 cases and, informed by this baseline of model performance, to systematically study the response of clouds to aerosol perturbations across all 54 cases."

We believe this new addition, along with the current explanations in the first paragraph of Sect. 4.1.1 about the first case and in the first paragraph of Sect. 4.2.1 about the second case (also mentioned below), should address the reviewer comment:

"Based on phase space analysis for satellite and reanalysis data (Fig. 4a), this case is characterized by an average PC2 value and a negative PC1 value. Among the 54 cases selected by PCA, and considering along-trajectory averages, it exhibits very strong 10-m WS, very weak $\omega$, nearly overcast conditions (~ 90%), and strong LWP (Fig. 4b)."

"The phase space analysis (Fig. 4a) shows that this case has the highest PC2 values and one of the lowest PC1 values among the 54 selected cases. Based on along-trajectory averages of physical variables (Fig. 4b), this case is characterized by very low $P_{MSL}$, very weak stability (lowest EIS), and high 700-hPa $q$."

● Mismatch in performance of the CTRL case between the two LES examples: While the authors do an excellent job discussing the impacts of varying Na within a given LES setup, I would like to see further explanation or analysis of the fact that the MERRA-2 diagnosed aerosol concentration is not a reliable choice to use for

initializing the model aerosol concentration.

- o Given that this work seeks to develop methods for analyzing MCB and SCT, the ability of the model to predict cloud behavior under different aerosol concentrations is of utmost importance. Considering that the LES cannot reliably predict the real cloud behavior using a renanalysis-derived value of Na, how much would you trust the model to predict a *hypothetical* brightened or thinned cloud, where there is no observation for comparison?

  To address this comment, we added a paragraph at the end of Sect. 5 (beginning at L897):

  "Given the absence of $N_a$ observations away from aircraft campaigns, significant uncertainty exists in our initial $N_a$ and will lead to uncertainty in the response of these cases to perturbed aerosols. When the initial $N_a$ is erroneous, we expect biased responses to aerosol perturbations, with stronger biases for precipitating clouds and weaker biases for non-precipitating clouds, as suggested by previous studies (e.g., Chun et al., 2023). However, since the LWP, LCC, and SW CRE of our unperturbed simulations depend both directly and indirectly on the initial Na, the agreement of these quantities with observations suggests that our initial $N_a$ is consistent with observations, within the limitations of our model. The GPCI S10 case exemplifies this, as comparisons indicate that MERRA Na is inconsistent with both observations and our model framework. While such an internal consistency check between unperturbed simulations and observations does not guarantee that the initial $N_a$ is correct, it strengthens confidence in our predicted responses to aerosol perturbations relative to a case without such a check."

- o In L895, you propose initializing with CERES Nd instead, but figures 8 and 9 both indicate (a) very poor temporal coverage; and (b) strong variability in the CERES value. How would you mitigate these issues to come up with a trustworthy value for your forward model?

  Discussion on deriving $N_a$ from CERES $N_d$ to initialize LES is beyond the scope of this study, and we will provide a detailed methodology in future publications. However, to address this comment, we note that the decreased CERES $N_d$ at the start and end of each day in the second case (Sandu 2010 trajectory) is not real but essentially an artifact of the high solar zenith angle. For this reason and to avoid confusion, we now show Nd time series for a time window with the zenith angle lower than 50° in Figs. 6 and 9. In addition, to account for natural temporal variability, we propose to use daytime averaging

when correcting $N_a$ in the future. While reliable nighttime $N_d$ is unavailable, our LES simulations indicate a negligible diurnal cycle of $N_d$. Therefore, the daytime-averaged $N_d$, corrected by a factor derived from an empirical relationship between $N_a$ and $N_d$ from the literature, should provide a reasonable estimate for initializing $N_a$ in the LES.

- If MERRA-2 offers a predicted value of Nd, it would be useful to show that against the CERES retrieved Nd and LES experiments. A close match between MERRA-2 and CERES Nd would indicate that the prognosed MERRA-2 Na is actually useful, but that some difference or deficiency in the aerosol-cloud parameterizations in the LES leads to the underprediction of Nd in certain cases.

We are not aware of any MERRA2 $N_d$ product, and it is not feasible for us to calculate $N_d$ from MERRA2 data, as we do not have access to all the required liquid cloud microphysical variables for the region and time period of this study (assuming MERRA2 provides such variables). Nonetheless, even if such a dataset were available, we would not expect it to agree well with observations. The Goddard Earth Observing System (GEOS) model used for MERRA-2 is not aerosol-interactive, meaning that aerosols do not affect cloud microphysics in the model. This further limits the ability of MERRA-2 for predicting $N_a$. In addition, reanalysis cloud products, such as cloud cover and liquid water path, often show poor agreement with satellite observations, as shown in Figs. 5 and 8 and explained in the 3rd paragraph in Sect. 5.

Further research is required to achieve a more accurate $N_a$ dataset on a global scale and for a long period of time. Assimilating satellite products with higher temporal resolution (e.g., Geostationary Operational Environmental Satellite or GOES) or incorporating satellites with vertical profile information (e.g., CALIPSO) could improve the accuracy of such datasets in the future.

We modified the 4th paragraph in Sect. 5 and added the above statement for more clarification (beginning at L881):

"Over the NEP and at higher values of $N_a$, MERRA2 $N_a$ is biased low when compared to in-situ measurements (Erfani et al., 2022). In the future, a critical step in forcing and initializing our LES with MBL aerosols will be to base aerosol concentrations on a combination of multiple observational (e.g., CERES $N_d$) and reanalysis (e.g., MERRA2 $N_a$) datasets, rather than using MERRA2 alone, to ensure more reliable simulations of ACIs. Given that MERRA2 $N_a$ is simulated by assimilating MODIS aerosol optical depth (which represents the optical property of aerosols throughout the column of troposphere), it can be

inaccurate at certain levels and locations, and is often subject to sudden changes associated with the aerosol optical depth data assimilation. Further research is required to achieve a more accurate $N_a$ dataset on a global scale and for a long period of time. Assimilating satellite products with higher temporal resolution (e.g., Geostationary Operational Environmental Satellite or GOES) or incorporating satellites with vertical profile information (e.g., CALIPSO) could improve the accuracy of such datasets in the future. For now, CERES provides satellite estimates of $N_d$ in the cloud layer, so it likely reflects $N_a$ more accurately in these Sc clouds. These estimates also result in $N_a$ values that are more consistent with other CERES products, such as TOA radiative fluxes, which are considered the most accurate measurements (Su et al., 2015)."

Minor Comments:

- L87-88: LES resolves some subgrid-scale turbulence that is not resolved in a GCM, but it does not actually represent aerosols or cloud microphysics any differently than most GCM approaches (moment and modal models), unless you are specifically referring to using a sectional or superdroplet approach.

We changed this sentence in Sect. 1 (beginning at L83):

"Large-eddy simulation (LES), on the other hand, proves more effective since it is able to resolve turbulence, convection, clouds, and precipitation within the marine boundary layer or MBL (Wyant et al., 1997; Sandu and Stevens, 2011; Berner et al., 2013; Blossey et al., 2021)"

- L116+: It would be helpful to clarify at the start of this paragraph that you are referring to the present study or the previous Erfani 2022 work.

We intend to briefly describe the progress done by the previous works that led to the current study. The beginning of this paragraph is modified to address the reviewer's comments (Sect. 1, beginning at L112):

"Previous studies used data from an observational field campaign to both initialize and then test the fidelity of one LES model (System for Atmospheric Modeling or SAM; see Sect. 2.4) in simulating the SCT in the northeast Pacific (NEP) region (Blossey et al., 2021; Mohrmann et al., 2019)..."

- Figure S1 is fuzzy / poor image quality

  We have changed both panels in Fig. S1 to improve their resolution and quality.

- L602: typo in "initiation"

  Done.

[Figure]

Figure R1. As in Fig. 3a, but with including SST and $P_{MSL}$ in PCA.

[Figure]

Figure R2. As in Fig. 4b, but the sampling is done randomly.

---

## Author Response (AR2)

**Response to the reviewer's comments on the manuscript**

Title: Building a comprehensive library of observed Lagrangian trajectories for testing modeled cloud evolution, aerosol-cloud interactions, and marine cloud brightening
By: Ehsan Erfani, Robert Wood, Peter Blossey, Sarah J. Doherty, Ryan Eastman
Article reference: egusphere-2024-3232

**REVIEWER COMMENTS:**

**Reviewer #1:**
**N/A**

**Reviewer #2:**
I am satisfied the authors' thoughtful and thorough response to comments and appreciative of their additional quantitative analysis. The additional description of trajectory package and additional justification of the use of PCA and exclusion of correlated variables is convincing. I feel that this article is suitable for publication and have only two minor suggestions which could further strengthen the article:

1. Include figure R2 in the supplement as well, instead of stating "Figure not shown." It is a highly convincing quantitative motivation for PCA.
We thank the reviewer for their constructive comments. To answer the above comment, we added Figure R2 to the supplement (now, Figure S2b) and modified the text to reflect this (by changing "figure not shown" to "Fig. S2b" in Line 514).

2. I still feel it is inaccurate to claim that "The two trajectories used for LES modeling were selected to represent distinct regions of the PC1-PC2 space", as both trajectories are in the upper left tail of an otherwise symmetric 2D distribution. I would remove this statement or reframe it as something like: "selected to illustrate distinct regimes among important cloud and atmospheric properties including…[P_MSL, CTH, LWP, etc.]"
We deleted this sentence and replaced it with the one suggested by the reviewer (Line 525).